# Molecular flexibility of DNA as a key determinant of RAD51 recruitment

Federico Paoletti[1], Afaf H El-Sagheer[2,3], Jun Allard[4], Tom Brown[2], Omer Dushek[1,*,†] (iD) & Fumiko Esashi[1,**,†] (iD)

## Abstract

**The timely activation of homologous recombination is essential for the maintenance of genome stability, in which the RAD51 recombinase plays a central role. Biochemically, human RAD51 polymerises faster on single-stranded DNA (ssDNA) compared to double-stranded DNA (dsDNA), raising a key conceptual question: how does it discriminate between them? In this study, we tackled this problem by systematically assessing RAD51 binding kinetics on ssDNA and dsDNA differing in length and flexibility using surface plasmon resonance. By directly fitting a mechanistic model to our experimental data, we demonstrate that the RAD51 polymerisation rate positively correlates with the flexibility of DNA. Once the RAD51-DNA complex is formed, however, RAD51 remains stably bound independent of DNA flexibility, but rapidly dissociates from flexible DNA when RAD51 self-association is perturbed. This model presents a new general framework suggesting that the flexibility of DNA, which may increase locally as a result of DNA damage, plays an important role in rapidly recruiting repair factors that multimerise at sites of DNA damage.**

**Keywords** double-stranded DNA; mathematical model; RAD51; single-stranded DNA; surface plasmon resonance

**Subject Categories** DNA Replication, Recombination & Repair; Structural Biology

The EMBO Journal (2020) 39: e103002

See also: **S Subramanyam & M Spies** (April 2020)

## Introduction

DNA double-strand breaks (DSBs) are cytotoxic lesions that can lead to chromosomal breaks, genomic instability and tumorigenesis in mammalian cells (Tubbs & Nussenzweig, 2017). Homologous recombination (HR) can offer an error-free DNA repair mechanism to restore genetic information at DSB sites and, in this way, contribute to genome stability. During HR-mediated repair, single-stranded DNA (ssDNA) overhangs are generated and rapidly coated with the ssDNA-binding replication protein A (RPA) (Chen & Wold, 2014). ssDNA-bound RPA is then exchanged for RAD51, the central ATP-dependent recombinase that catalyses HR-mediated repair. RAD51 polymerises on ssDNA to form a nucleoprotein filament and guide homologous strand invasion and DSB repair (Baumann *et al*, 1996).

The central mechanism of HR is evolutionarily highly conserved, and the bacterial RAD51 ortholog RecA has clear preference to polymerise on ssDNA over dsDNA (Benson *et al*, 1994). However, human RAD51 shows weaker binding affinity to ssDNA compared to RecA, and the mechanism by which RAD51 polymerises on ssDNA in preference to dsDNA remains enigmatic. Earlier studies using electrophoretic mobility shift assay (EMSA) have suggested that RAD51 binds both ssDNA and dsDNA with similar affinities (Benson *et al*, 1994), albeit its preferential ssDNA binding was visible in the presence of ammonium sulphate (Shim *et al*, 2006). These endpoint assays, however, do not provide information as to whether binding kinetics may contribute to a potential RAD51-dependent ssDNA/dsDNA discrimination mechanism. Indeed, more recent kinetic studies have revealed that RAD51 polymerisation consists of two phases: a rate-limiting nucleation phase and a growth phase (Miné *et al*, 2007; Van der Heijden *et al*, 2007; Hilario *et al*, 2009). A minimal polymer nucleus, with a length of either two-to-three or four-to-five RAD51 protomers (Van der Heijden *et al*, 2007; Hilario *et al*, 2009; Subramanyam *et al*, 2016), is proposed to elicit the growth phase of RAD51 polymerisation. Intriguingly, RAD51 was shown to display faster association kinetics on ssDNA (Candelli *et al*, 2014) and slower dissociation kinetics on dsDNA, indicating that the RAD51-dsDNA complex is stable once formed (Miné *et al*, 2007). However, the mechanism underlying these kinetic differences is ill-defined.

A key difference between ssDNA and dsDNA is their molecular flexibility: ssDNA is known to be more flexible compared to dsDNA.

1 Sir William Dunn School of Pathology, University of Oxford, Oxford, UK
2 Department of Chemistry, University of Oxford, Oxford, UK
3 Department of Science and Mathematics, Suez University, Suez, Egypt
4 Department of Mathematics, University of California, Irvine, CA, USA
*Corresponding author. Tel: +44 1865 275576; E-mail: omer.dushek@path.ox.ac.uk
**Corresponding author. Tel: +44 1865 275289; E-mail: fumiko.esashi@path.ox.ac.uk
†These authors contributed equally to this work

The flexibility of a DNA molecule can be characterised by its persistence length ($L_p$), a mechanical parameter quantifying polymer rigidity: the higher the persistence length, the more rigid the polymer. In the presence of monovalent or divalent salt, dsDNA displays an $L_p$ of ~ 30–55 nm (Baumann *et al*, 1997; Brunet *et al*, 2015), while ssDNA is much more flexible, with an $L_p$ of 1.5–3 nm (Murphy *et al*, 2004; Chi *et al*, 2013; Kang *et al*, 2014). These observations imply that ssDNA can explore a much larger configurational space compared to dsDNA. It follows that the formation of a structured (less flexible) RAD51 polymer on ssDNA will need to offset a greater deal of ssDNA's configurational freedom, referred to as entropic energy, compared to the formation of a RAD51 polymer on dsDNA. Despite the clear thermodynamic implications of RAD51 polymerisation on DNA, the impact of DNA flexibility on RAD51 nucleoprotein filament formation has been largely overlooked.

In this study, we describe how RAD51 polymerises on DNA. Using a combination of surface plasmon resonance (SPR) and small-angle X-ray scattering (SAXS), we have assessed the RAD51 binding kinetics on DNA using ssDNA and dsDNA oligonucleotides differing in length and flexibility. Analyses of the SPR data using biochemical mathematical models revealed that RAD51 polymerisation required a minimal nucleus of four and two molecules on ssDNA and dsDNA, respectively. Interestingly, our analyses further uncovered that RAD51 is a mechano-sensor, a biomolecule that polymerises faster on more flexible DNA. This is surprising, because polymerisation on more flexible DNA should produce less stable polymers due to a larger configurational confinement. Therefore, we hypothesised that this confinement cost, defined as entropic penalty, is offset by a strong RAD51 protomer–protomer interaction and show this to be the case by analysing a RAD51 point mutant which is defective in self-association. We propose that RAD51 sensitively recognises locally flexible DNA, which may be generated at sites of DNA damage, and so rapidly forms nucleoprotein filament for HR repair.

# Results

## RAD51 preferential binding to ssDNA is dependent on the length of the DNA template

To evaluate how human RAD51 discriminates between ssDNA and dsDNA, we used SPR to assess the binding kinetics of untagged recombinant human wild-type (WT) RAD51 to a 50-mer mixed-base ssDNA molecule (dN-50) and a 50-mer mixed-base paired dsDNA molecule (dN-50p) (Table 1). These analyses, conducted in the presence of ATP and $Ca^{2+}$ to block RAD51 ATP hydrolysis (Bugreev & Mazin, 2004), showed (i) faster RAD51 association with ssDNA compared to dsDNA, and (ii) similar RAD51 lifetimes on both ssDNA and dsDNA (Fig 1A and B). These observations confirm previous findings that WT RAD51 distinguishes ssDNA from dsDNA through faster polymerisation, but not the stability of polymers, on ssDNA (Candelli *et al*, 2014), validating SPR as a sensitive experimental system with which to determine the kinetics of RAD51 binding to DNA.

We then aimed to define the initial steps of RAD51 association with ssDNA. To this end, we generated a series of mixed-base short ssDNA oligonucleotides with varying lengths, each of which can be bound by a restricted number of RAD51 molecules. As a single RAD51 molecule engages with three nucleotides of DNA (Short *et al*, 2016), a ssDNA consisting of five nucleotides (dN-5), eight nucleotides (dN-8), 11 nucleotides (dN-11), 14 nucleotides (dN-14) and 17 nucleotides (dN-17) can accommodate up to one, two, three, four and five RAD51 molecules, respectively (Fig 2A and Table 1). Our systematic SPR measurements of WT RAD51 binding kinetics revealed no RAD51 binding to the dN-5, dN-8 and dN-11, slow association and moderate dissociation with dN-14, and slightly faster association and slow dissociation with dN-17 (Fig 2A and B). These observations suggested that three or fewer RAD51 molecules are unable to generate a stable nucleus on ssDNA, four RAD51

**Table 1.** The list of DNA oligonucleotides used in this study.

| Name | Mass (kDa) | Sequence |
|---|---|---|
| ssDNA dN-5 | 2.057 | 5′-CGGAC-Biotin TEG-3′ |
| dsDNA dN-5p | 4.286 | 5′-CGGAC LL GTCCG-Biotin TEG-3′ |
| ssDNA dN-8 | 2.594 | 5′-CTGACTGC- Biotin TEG-3′ |
| dsDNA dN-8p | 6.139 | 5′-CTGACTGC LL GCAGTCAG-Biotin TEG-3′ |
| ssDNA dN-11 | 3.926 | 5′-CGTCGATAGGC-Biotin TEG-3′ |
| dsDNA dN-11p | 7.993 | 5′-CGTCGATAGGC LL GCCTATCGACG-Biotin TEG-3′ |
| ssDNA dN-14 | 4.872 | 5′-CGTCGATGAGCAGT-Biotin TEG-3′ |
| ssDNA dN-17 | 5.795 | 5′-CGTCGATGAGCAGTGTC-Biotin TEG-3′ |
| ssDNA dN-50 | 15.682 | 5′-Biotin-TTGAGAGAGCAGACCACAATTATCACCTACACGACATCATTTTATATCAA-3′ |
| dsDNA dN-50p | 31.157 | (Forward) 5′-Biotin-TCGAGAGGGTAAACCACAATTATCGCCTACCCAAAACTATTTTATATCAA-3′ |
| | | (Reverse) 5′-TTGATATAAAATAGTTTGGGTAGGCGATAATTGTGGTTTACCCTCTCGA-3′ |
| ssDNA dT-50 | 15.541 | 5′-Biotin-TTTTTTTTTTTTTTTTTTTTTTTTTTTTTTTTTTTTTTTTTTTTTTTTTT-3′ |
| ssDNA dA-50 | 15.992 | 5′-Biotin-AAAAAAAAAAAAAAAAAAAAAAAAAAAAAAAAAAAAAAAAAAAAAAAAAA-3′ |

Biotin TEG: biotin linked to a flexible triethylene glycol spacer; Biotin: biotin linked to a standard C6 spacer; dA: deoxyadenosine; dN: mixed-base composition; dT: deoxythymidine; L: hexaethylene glycol flexible linker.

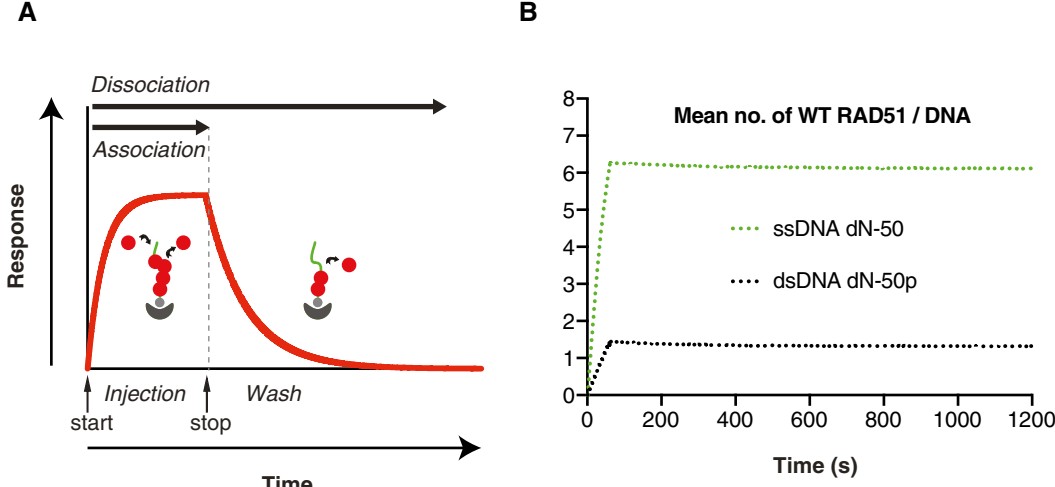

**Figure 1. The experimental setup.**

A  Depiction of the experimental setting. A biotinylated DNA oligonucleotide is immobilised onto a surface plasmon resonance (SPR) CM5 chip via biotin–streptavidin interaction. RAD51 protein is injected over the DNA-coated SPR matrix to measure polymerisation kinetics. Throughout this period, association and dissociation of RAD51 take place simultaneously. Following protein injection (stop), running buffer is injected to measure dissociation kinetics.

B  Wild-type (WT) RAD51 SPR curves for ssDNA dN-50 and dsDNA dN-50p. RAD51 was injected at 150 nM in the presence of 2.5 mM ATP (pH 7.5) and 10 mM CaCl$_2$. The normalised mean number of WT RAD51 bound to respective DNA oligonucleotides ($N$) is plotted versus time, as measured by SPR, following the equation $N = S/(L * (M_{RAD51}/M_{DNA}))$, where $S$ is the signal in RU units (1 RU ~ 50 pg/mm$^2$), $L$ is the amount of DNA ligand immobilised onto the experimental flow cell (RU), $M_{RAD51}$ is the molecular weight (kDa) of RAD51 (~ 37 kDa) and $M_{DNA}$ is the molecular weight (kDa) of the immobilised DNA molecule.

Source data are available online for this figure.

molecules form a quasi-stable nucleus, and five or more RAD51 molecules are able to form a highly stable nucleus.

To similarly evaluate the initial steps of RAD51 association with dsDNA, we generated an analogous series of mixed-base dsDNA consists of five base pairs (dN-5p), eight base pairs (dN-8p) and 11 base pairs (dN-11p), which can accommodate up to one, two or three RAD51 molecules, respectively (Fig 2C and Table 1). Our SPR measurements detected no RAD51 binding to the dsDNA dN-5p, as was the case for the ssDNA dN-5. To our surprise, however, we detected slow association and moderate dissociation of WT RAD51 with the dN-8p, and even faster association and slower dissociation with the dN-11p (Fig 2D). These observations suggested that, while RAD51 molecule is unable to associate stably with dsDNA, two RAD51 molecules form a temporary stable (quasi-stable) nucleus and three or more RAD51 molecules are able to form a highly stable nucleus. This suggested that WT RAD51 nucleation on dsDNA requires only two-to-three RAD51 molecules versus four-to-five on ssDNA. Altogether, these observations support the idea that WT RAD51 can bind more strongly to short (≤ 11 bp) dsDNA compared to short (≤ 11 nt) ssDNA molecules, but binds more strongly to long (50 nt) ssDNA molecules compared to long (50 bp) dsDNA molecules.

**RAD51 polymerisation on ssDNA is promoted by faster adsorption and/or elongation compared to dsDNA**

Informed by these SPR datasets, we developed an ordinary differential equation (ODE) model with three modules: RAD51 polymerisation in solution, RAD51 polymerisation on ssDNA and RAD51 polymerisation on dsDNA. This model was globally fitted

to all corresponding SPR curves in order to determine the mechanistic differences in RAD51 polymerisation kinetics on ssDNA and dsDNA.

The module of RAD51 polymers in solution was modelled on the basis of mass action kinetics with a maximum length of 16 (see Appendix Methods and Appendix Table S1 for details). The model allows any RAD51 $n$-mer ($1 \leq n < 16$) to associate with any other RAD51 $m$-mer ($1 \leq m < 16$) to form a $(m + n)$-mer ($1 < m + n \leq 16$). In addition, any RAD51 $k$-mer can fall apart in every possible combination of $m$-mers and $n$-mers ($k = n + m$; e.g. a pentamer can fall apart to form a monomer and a tetramer, or a dimer and a trimer). This model provides the concentrations of RAD51 polymers in solution and is solved in the steady-state so that it only depends on a single fit parameter $K_D$ (i.e. the RAD51 protomer–protomer dissociation constant). The steady-state assumption is reasonable because in SPR there is a constant flow replenishing any RAD51 that binds to the surface.

The two kinetic modules of RAD51 polymer formation on ssDNA and dsDNA were identified by increasing complexity based on mass action kinetics until we identified models that were able to fit the experimental datasets (see Appendix Methods, Appendix Fig S1 and Appendix Table S2 for details). Both the ssDNA and dsDNA modules allow for the adsorption of any RAD51 $n$-mer from solution onto DNA (provided at least $n$ DNA-binding sites are available), given that $n$-mers ($1 \leq n \leq 16$) can exist in solution. Polymer elongation occurs when a RAD51 $m$-mer in solution binds to a DNA-bound $n$-mer to form a DNA-bound $(n + m)$-mer, provided at least $n + m$ DNA-binding sites are available and $1 < m + n \leq 16$. However, unbinding was assumed to take place only via single

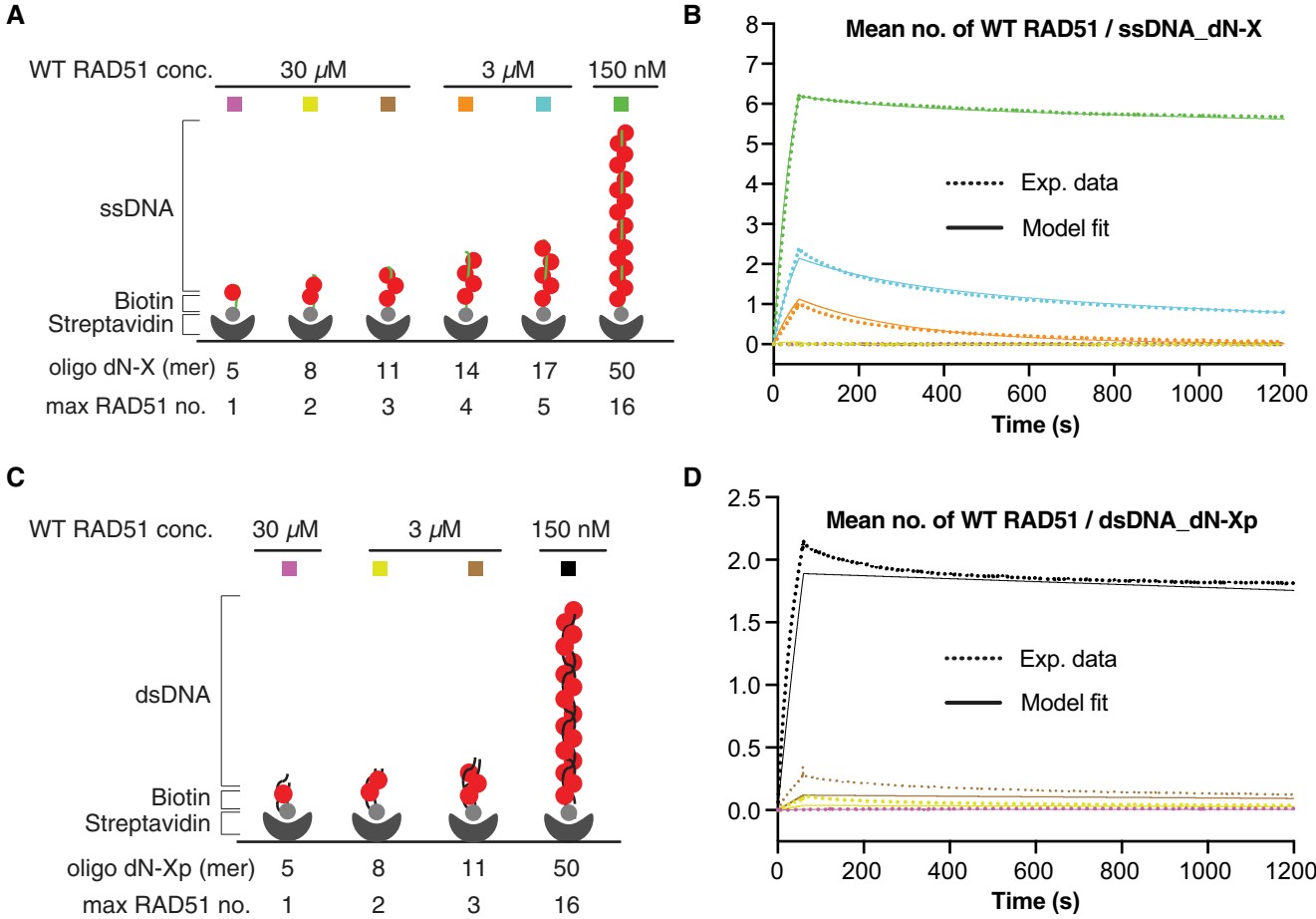

**Figure 2.  Kinetics of WT RAD51 binding to ssDNA and dsDNA oligonucleotides of varying length.**

A   Biotinylated ssDNA oligonucleotides of indicated lengths were separately immobilised onto SPR CM5 chips via biotin–streptavidin interaction. Wild-type (WT) RAD51 was injected at the indicated concentrations to measure association and dissociation kinetics.

B   The dotted and solid curves show the normalised mean number of RAD51 bound to the ssDNA oligonucleotides, as measured by SPR (see Fig 1 for equation) and the ODE model fits (see Fig 3A for model description), respectively.

C   As in (A), except biotinylated dsDNA molecules of indicated lengths were used to measure association and dissociation kinetics.

D   As in (B), the dotted and solid curves show the mean number of RAD51 bound to the dsDNA oligonucleotides, as measured by SPR and the ODE model fits (see Fig 3B for model description), respectively.

Source data are available online for this figure.

protomer dissociation and via the dissociation of short RAD51 nuclei. This is a valid assumption considering that, in all the SPR experiments in this study, ATP hydrolysis was inhibited due to the presence of $Ca^{2+}$. This condition would lead to slow RAD51 filament disassembly which can be approximated as disassembly via monomer removal or via the removal of short, unstable RAD51 nuclei. The ssDNA module consists of a polymerisation forward rate constant ($k_p$), an unstable reverse rate constant ($k_u$), a quasi-stable reverse rate constant ($k_q$) and a stable reverse rate constant ($k_s$). $k_p$ describes the adsorption and elongation of RAD51 polymers on ssDNA up to a maximum length of 16, $k_u$ describes the dissociation of unstable nuclei (one to three RAD51 molecules), $k_q$ describes the dissociation of quasi-stable nuclei (four RAD51 molecules), and $k_s$ describes the dissociation of single RAD51 protomers (i.e. RAD51 monomers of a RAD51 polymer; Figs 2A and 3A). Similarly, the kinetic module for dsDNA includes $k_p$, $k_u$ and $k_s$, but without the

need for $k_q$ to describe dissociation of quasi-stable nuclei (i.e. two RAD51 molecules; Fig 2C). For dsDNA, $k_u$ describes the rate of dissociation of RAD51 monomers not bound to a RAD51 polymer (Fig 3B).

Using ABC-SMC to fit the aforementioned modules of RAD51 polymers in solution and on DNA to the experimental data of ssDNA dN-X (Fig 2B) and dsDNA dN-Xp (Fig 2D), values for the model parameters were determined (Figs 3A and B, and EV1). This analysis identified a nano-molar range RAD51 protomer–protomer dissociation constant (dN-X and dN-Xp $K_D = 1.14 \pm 0.5$ nM) as a key factor driving rapid RAD51 polymerisation on both ssDNA and dsDNA, similarly to other studies using either pressure perturbation fluorescence spectroscopy (Schay *et al*, 2016) or single molecule fluorescence microscopy (Candelli *et al*, 2014). This is due to the fact that this $K_D$ value enables WT RAD51 to form abundant long polymers in solution at both 150 nM and 3 μM [WT RAD51]

(Fig 3C and D). These RAD51 polymers can then directly adsorb onto DNA and elongate, therefore enabling WT RAD51 to polymerise faster on the dN-50 and dN-50p (150 nM [WT RAD51]) compared to the dN-14, dN-17, dN-11p and dN-8p oligonucleotides (3 μM [WT RAD51]) despite a 20-fold lower RAD51 injection concentration.

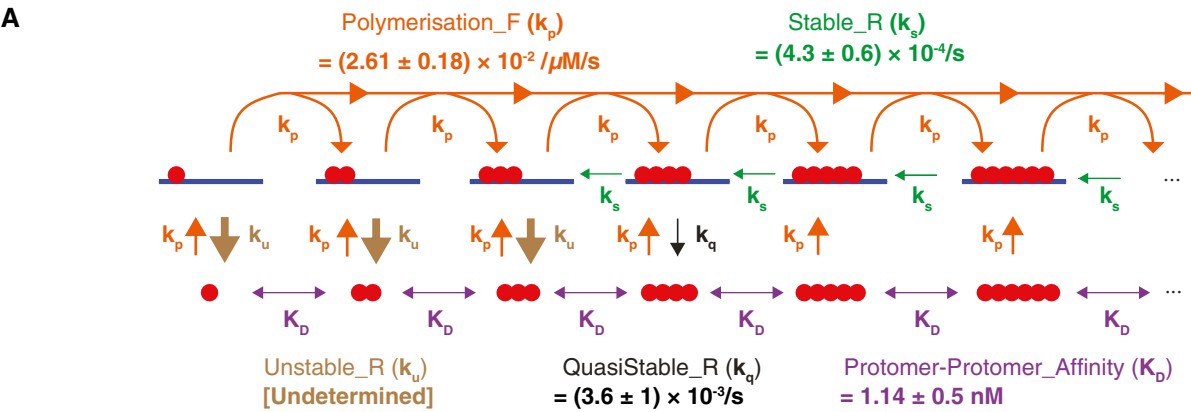

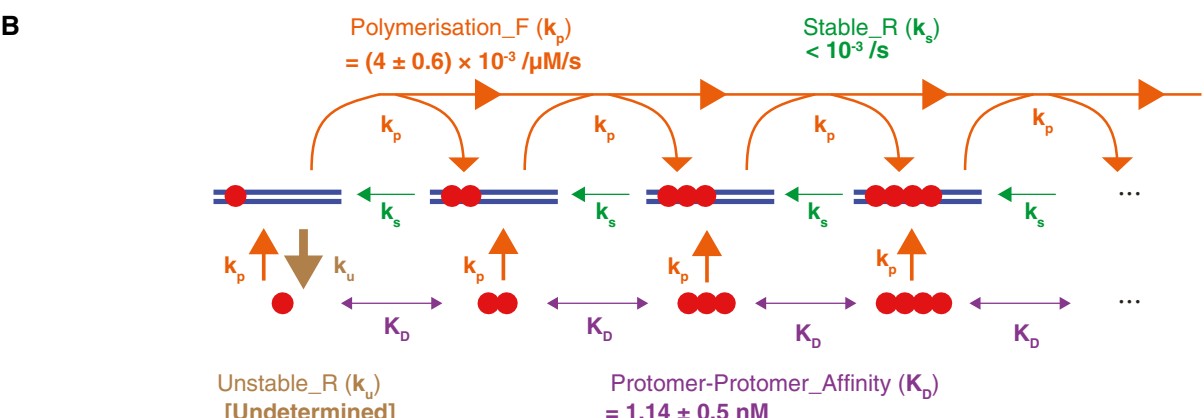

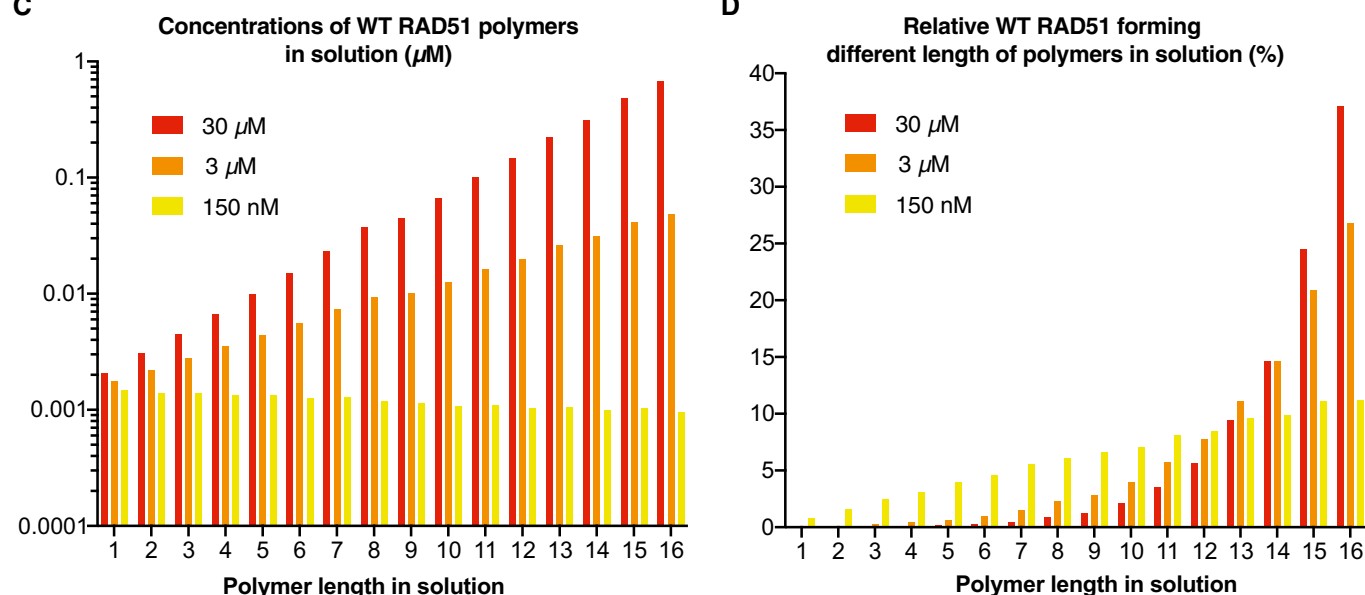

**Figure 3.**

◀

**Figure 3.  Mathematical models describing WT RAD51 polymerisation kinetics on ssDNA and dsDNA.**

A, B   (A) Kinetic representation of the ordinary differential equation (ODE) model describing WT RAD51 polymerisation on ssDNA, consisting of five parameters: $k_p$ (polymerisation forward rate), $k_u$ (unstable reverse rate), $k_q$ (quasi-stable reverse rate), $k_s$ (stable reverse rate) and $K_D$ (protomer–protomer interaction affinity). $K_D$ predicts the concentrations of RAD51 polymers of variable length in solution, while $k_p$, $k_u$, $k_q$ and $k_s$ predict the speed of formation of RAD51 polymers on ssDNA. (B) Kinetic representation of the ODE model describing WT RAD51 polymerisation on dsDNA, consisting of four parameters: $k_p$, $k_u$, $k_s$ and $K_D$. $K_D$ predicts the concentrations of RAD51 polymers of variable length in solution, and $k_p$, $k_u$ and $k_s$ predict the speed of formation of RAD51 polymers on dsDNA. In panels (A, B), purple arrows depict a simplified cartoon representation of RAD51 polymerisation in solution. The model allows for any RAD51 $n$-mer ($1 \leq n < 16$) to associate with any other RAD51 $m$-mer ($1 \leq m < 16$) to form an ($m + n$)-mer ($1 < m + n \leq 16$), and any RAD51 $k$-mer to dissociate into any combination of $n$-mers and $m$-mers ($k = n + m$). The ssDNA and dsDNA models were calibrated by simultaneously fitting the SPR curves of ssDNA dN-X (dN-8, dN-14, dN-17, dN-50) and dsDNA dN-Xp (dN-5p, dN-8p, dN-11p, dN-50p) using the mode ABC-SMC particles. Mean values ± 1 SD of three mode ABC-SMC particles derived from model fits of three dN-X and dN-Xp repeats ($n = 3$). ssDNA $k_p$, $k_u$, $k_q$ and $k_s$ were fit to the ssDNA SPR curves, and dsDNA $k_p$, $k_u$ and $k_s$ were fit to the dsDNA SPR curves. A single $K_D$ was fit to all curves.

C, D   (C) Predicted concentrations of WT RAD51 polymers in solution at equilibrium as a function of WT RAD51 monomer concentration (150 nM, 3 μM, 30 μM), prior to RAD51 injection onto DNA-coated SPR CM5 chips. (D) Predicted % WT RAD51 within each polymer state in solution at equilibrium as a function of WT RAD51 monomer concentration (150 nM, 3 μM, 30 μM), prior to RAD51 injection onto DNA-coated SPR CM5 chips. In panels (C, D), the $K_D$ was fixed to the mean value identified in Fig 3A and B (i.e. $K_D = 1.14$ nM), and the concentration of WT RAD51 monomer concentration was varied accordingly. In panel (D), the % WT RAD51 values were calculated by multiplying the polymer concentrations by their respective polymer length and dividing each value by the total RAD51 monomer concentration (i.e. % WT RAD51 = [$n$-mer] * $n$/[WT RAD51$_{monomer}$]). [WT RAD51$_{monomer}$] was calculated via Coomassie staining image quantification using a BSA standard curve. It is important to note that the assumed maximum polymer length in solution (16-mer) leads to an overestimation of the concentration of each $n$-mer in solution, given that RAD51 is likely to form polymers of length greater than 16 at 150 nM, 3 μM and 30 μM WT RAD51 monomer concentration. However, this overestimation is likely to not alter the conclusion that WT RAD51 forms long polymers in solution at 150 nM–30 μM concentration.

Source data are available online for this figure.

Importantly, we also identified a 6.5-fold higher ssDNA polymerisation forward rate constant (dN-X $k_p = (2.6 \pm 0.18) \times 10^{-2}$/μM/s) compared to dsDNA (dN-Xp $k_p = (4 \pm 0.6) \times 10^{-3}$/μM/s), explaining the overall faster RAD51 polymerisation on ssDNA (Fig 3A and B). Together, these analyses suggest that WT RAD51 adsorption and/or elongation is faster on ssDNA compared to dsDNA, and that a high protomer–protomer affinity enables WT RAD51 to nucleate and elongate effectively on ssDNA and dsDNA even at low concentrations of RAD51.

## RAD51 polymerises faster on flexible DNA

Overall, our analyses suggest that WT RAD51 can nucleate more efficiently on short dsDNA molecules but polymerises significantly faster on long ssDNA molecules. We speculate that the difference in RAD51 polymerisation speed is due to the higher flexibility of ssDNA compared to dsDNA. Explicitly, we propose a Bend-To-Capture (BTC) mechanism to explain how DNA flexibility can impact polymerisation kinetics (Fig 4A). A free RAD51 monomer or polymer in solution would need to generate two sequential, non-covalent interactions to be incorporated into the growing polymer: the interaction with the exposed interface of an existing RAD51 polymer and with the exposed scaffold DNA. In order for the incoming RAD51 to fit into the position without steric clashes, we reasoned that naked DNA immediately next to the existing RAD51 polymer needs to be in a configuration that can bend away from the preferred direction of the polymer. In this way, flexible DNA is expected to explore more conformations compatible with the further addition of RAD51 per unit time.

To test this notion, we designed an experiment to measure the kinetics of WT RAD51 binding to DNA oligonucleotides of varying flexibility. It has been shown that poly-dT ssDNA, which is widely used for RAD51 binding assays, is highly flexible, while poly-dA ssDNA is highly rigid due to base stacking interactions (Mills *et al*, 1999; Sim *et al*, 2012). Consistently, our SAXS-derived persistence length ($L_p$) measurements showed ssDNA dT-50 is the most flexible oligonucleotide, followed by dN-50, dA-50 and dsDNA dN-50p (Fig EV2 and Table 2). By measuring RAD51 binding kinetics to these

DNA oligonucleotides (Fig 4B and Table 1), we found that WT RAD51 indeed displayed faster association with the dT-50 compared to the dN-50, and the model fit suggests this is due to a higher polymerisation rate constant ($k_p$; Fig 4C and D). Furthermore, WT RAD51 displayed very slow association with dA-50, comparable to that of dsDNA dN-50p, which is explained by a lower polymerisation rate constant (Fig 4C and D). The difference is unlikely attributed to the preferential RAD51 association with thymine bases, as RAD51 binds to the sugar-phosphate backbone of ssDNA, exposing the nucleotide bases for homologous pairing (Short *et al*, 2016; Xu *et al*, 2017). Interestingly, the increase in RAD51 polymerisation rate was not linear beyond the $L_p$ of DNA at around ~ 4.7 nm (Fig 4D, Table 2), which is roughly on the same size scale as a single RAD51 molecule. This observation agrees with the BTC model that a single RAD51 can be added to an existing RAD51 polymer only on DNA of $L_p < 4.7$ nm, which can bend out of the way to allow RAD51 polymerisation. The ssDNA dT-50 or dA-50 data, however, could not be explained by individually fitting the unstable reverse rate constant ($k_u$), the quasi-stable reverse rate constant ($k_q$) or the stable reverse rate constant ($k_s$; Fig EV3A–C). In addition, for RAD51 DNA-bound polymers of length greater than three, the fractions of RAD51 in each polymer state are similar on ssDNA dT-50, dN-50, dA-50 and dsDNA dN-50p, suggesting WT RAD51 forms a similar proportion of polymers length 4–16 across the four DNA molecules (Fig EV3D). Finally, we observed that the lifetime of DNA-bound RAD51 protomers was independent of the flexibility of the underlying DNA (Fig 4E), which is in agreement with a previous study using optical tweezers (Brouwer *et al*, 2018). Taken together, these observations suggest that although WT RAD51 is able to form a stable nucleus with fewer molecules on dsDNA compared to ssDNA (two versus five), it elongates and/or adsorbs more efficiently on ssDNA because of its higher flexibility, consistent with our proposed BTC mechanism.

## The RAD51 protomer–protomer interaction is required for stable binding on flexible DNA

So far, we have observed that WT RAD51 associates faster with flexible ssDNA than with dsDNA, and the proposed BTC mechanism

*Federico Paoletti et al*

*The EMBO Journal*

explains how higher DNA flexibility can induce faster RAD51 polymerisation. However, the lifetime of WT RAD51 forming stiff polymers on DNA, $L_p$ of which is at ~ 200 nm for both RAD51-ssDNA and RAD51-dsDNA complexes (Miné *et al*, 2007), appeared to be independent of DNA flexibility (Fig 4E) (Brouwer *et al*, 2018). This is surprising because formation of stiff RAD51 polymers on more flexible DNA would incur a larger entropic penalty (Fig 4B, $\Delta TS_{DNA}$ [$L_p$]). To explain this apparent inconsistency, we hypothesised that

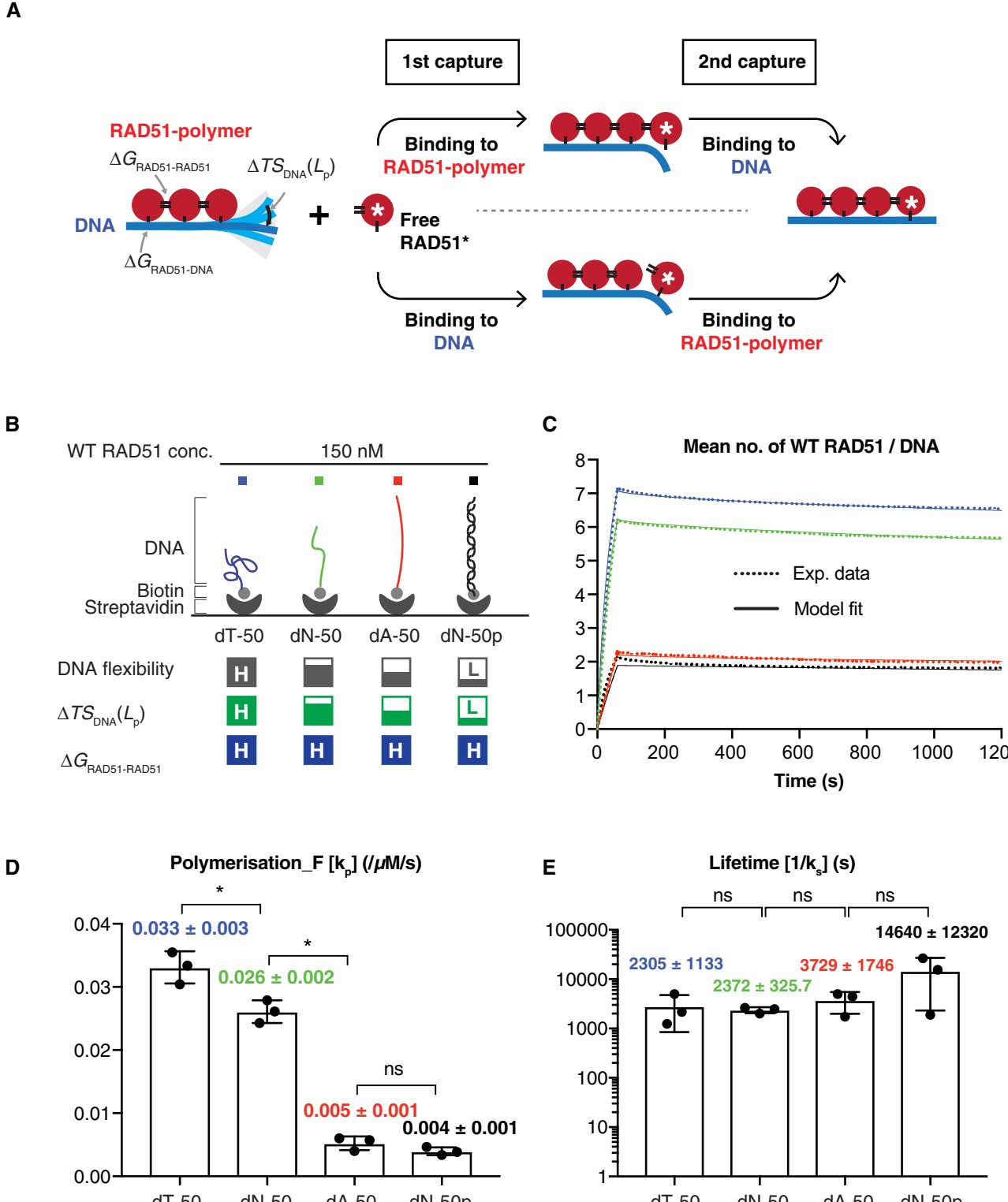

**Figure 4.**

◀

**Figure 4.   Kinetics of WT RAD51 binding to DNA of varying flexibility.**

A     A depiction of the Bend-to-Capture (BTC) mechanism. RAD51 generates two sequential, non-covalent interactions (a RAD51 protomer–protomer interaction and a RAD51-DNA interaction, or vice versa) faster on flexible DNA (depicted as ssDNA) compared to rigid DNA (depicted as dsDNA). The free RAD51 monomer or polymer in solution (here depicted as a monomer) to be incorporated into the growing polymer is shown with an asterisk. $\Delta TS$ is the entropic penalty of restricting DNA bending fluctuations, which depends on DNA persistence length. $\Delta G_{\text{RAD51-RAD51}}$ and $\Delta G_{\text{RAD51-DNA}}$ are binding energies, which do not depend on DNA persistence length, i.e. degrees of freedom within the macrostate of being bound or unbound.

B     Biotinylated DNA molecules of varying flexibility were separately immobilised onto SPR CM5 chips via biotin–streptavidin interaction, and WT RAD51 was injected at the indicated concentrations to measure association and dissociation kinetics. The flexibility of respective DNA, the expected entropic penalties upon RAD51 binding to corresponding DNA and protomer–protomer binding energy contribution of WT RAD51 are indicated in grey, green and blue boxes. H and L in each box denote high and low, respectively.

C     The mean number of WT RAD51 bound to respective DNA oligonucleotides, as measured by SPR (dotted lines) and ODE model fits (solid lines).

D, E   Bar plots of fitted $k_p$ values (D) and $k_s$ values (E) for each SPR curves. In panel (D), all parameters except $k_p$ were fixed to the mean values as identified in Fig 3A and B, and $k_p$ was fitted using lsqcurvefit (MATLAB). In panel (E), all parameters except $k_s$ and $k_p$ were fixed to the mean values as identified in Fig 3A and B, and $k_s$ and $k_p$ were fitted using lsqcurvefit (MATLAB). Only the $k_s$ values are reported here. Mean ± 1 SD of three ODE model fits ($n = 3$). (D): un-paired, one-tailed Mann–Whitney–Wilcoxon tests. (E): un-paired, two-tailed Mann–Whitney–Wilcoxon tests. *$P \leq 0.05$; ns, non-significant.

Source data are available online for this figure.

**Table 2.   Persistence lengths ($L_p$) ± 95% confidence intervals are estimated from fitted Kuhn lengths for the ssDNA dN-50, dA-50, dT-50 and the dsDNA dN-50p SAXS plots.**

| Oligonucleotide name | Kuhn length (nm) | Persistence length (nm) | Persistence length error (95% CI) (nm) |
|---|---|---|---|
| ssDNA dT-50 | 2.6702 | 1.3351 | 0.200185 |
| ssDNA dN-50 | 4.5177 | 2.25885 | 0.048095 |
| ssDNA dA-50 | 9.3488 | 4.6744 | 0.13126 |
| dsDNA dN-50p | 14.674 | 7.337 | 0.07379 |

Confidence interval (CI) for standard deviation. All data manipulation and model fitting were done using SasView. It is likely that the relatively low dsDNA dN-50p $L_p$ of 7.34 nm compared to the average dsDNA $L_p$ of 30–55 nm (Baumann *et al*, 1997; Brunet *et al*, 2015) is due to the low GC content of the dN-50p (36%).

the change of energy incurred by the new incorporation of WT RAD51 into an existing RAD51 filament (Fig 4B, $\Delta G_{\text{RAD51-RAD51}}$ and $\Delta G_{\text{RAD51-DNA}}$), which takes place independently of DNA bending and is referred to as enthalpic gain, is sufficiently large to dominate the entropic penalty. This model, defined as the Entropic Penalty Compensation (EPC) mechanism, proposes that the favourable binding energy of RAD51 polymerisation on DNA is sufficient to compensate for any DNA-dependent entropic penalty. If the binding energy between RAD51 protomers were reduced, our model predicts that, at equilibrium, there would be less RAD51 bound to flexible DNA compared to stiff DNA; hence, RAD51 would exhibit faster unbinding from flexible DNA compared to stiff DNA (Fig 5A).

To test this model directly, we assessed the kinetics of a RAD51 mutant, which confers a reduced protomer–protomer binding energy contribution compared to WT RAD51. We envisaged that, with such a mutation, the stability of the mutant RAD51 polymers on DNA would become more dependent on the flexibility of the underlying DNA (Fig 5A and B). We took advantage of a phenylalanine to glutamate substitution at RAD51 residue 86 (F86E), which reduces the RAD51 protomer–protomer interaction affinity without affecting the overall structure of the RAD51 molecule (Pellegrini *et al*, 2002; Yu *et al*, 2003; Esashi *et al*, 2007). Indeed, using size-exclusion chromatography with multi-angle light scattering (SEC-MALS), we confirmed that F86E RAD51 is primarily monomeric in solution (Fig EV4A–C). We then used SPR to measure the binding kinetics of F86E RAD51 to ssDNA dT-50, dN-50, dA-50 and dsDNA dN-50p (Fig 5C), and used ABC-SMC to fit a simplified polymerisation model simultaneously to all four datasets (Fig 5D). Importantly, we fitted a single reverse rate ($k_{\text{pr}} = k_u = k_q = k_s$) to the F86E RAD51 SPR data, given that the F86E RAD51 binding kinetics

to the short ssDNA oligonucleotides (dN-5, dN-8, dN-11, dN-14, dN-17) and dsDNA oligonucleotides (dN-5p, dN-8p, dN-11p) were not systematically measurable due to low F86E purification yield. It was immediately evident that F86E RAD51 shows significantly lower affinity for ssDNA and dsDNA compared to WT RAD51, with no detectable binding to DNA at the concentration of 150 nM or 3 μM (Fig EV4D and E). Nonetheless, at 30 μM, we observed that F86E RAD51 displays faster elongation on more flexible DNA (Figs 5B and E, and EV5), as we observed for WT RAD51 (Fig 4B and D). Additionally, DNA binding at 30 μM can be explained by the fact that F86E RAD51 ($K_D = 10.2 \pm 7$ μM) only forms abundant, nucleus-size polymers in solution (two-to-five RAD51 molecules) at 30 μM [F86E RAD51] (Fig 5F and G). This is in contrast to WT RAD51 ($K_D = 1.14 \pm 0.5$ nM), which can form abundant, long polymers in solution at 150 nM or 3 μM [WT RAD51] (Fig 3C and D). Strikingly, F86E RAD51 displayed reduced lifetimes ($1/k_{\text{pr}}$) that inversely correlated with the flexibility of DNA (Fig 5H). This observation is in sharp contrast to WT RAD51, which formed stable, long-lived polymers independently of DNA flexibility (Fig 4E). Taken together, these observations support the notion that the energetic contribution of WT RAD51 polymer formation is sufficient to offset the large entropic penalty associated with polymerisation on flexible DNA. As a result, RAD51 is able to polymerise more rapidly on flexible DNA despite incurring a larger entropic penalty.

## Discussion

In this study, we have analysed the kinetics of RAD51 binding to ssDNA and dsDNA oligonucleotides of varying length and flexibility

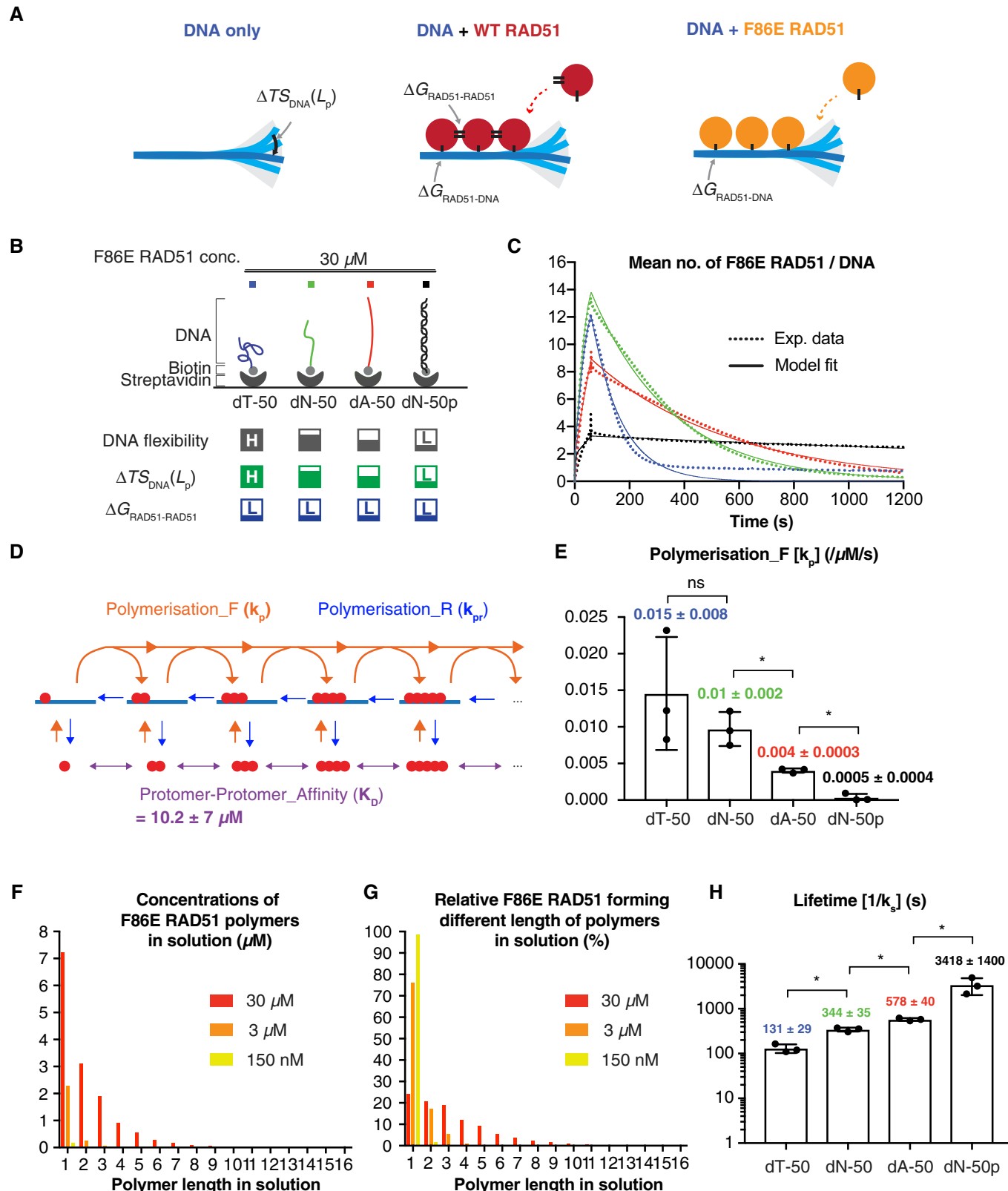

**Figure 5.**

**Figure 5.  Kinetics of monomeric F86E RAD51 binding to DNA of varying flexibility.**

A   A depiction of the Entropic Penalty Compensation (E/CPC) mechanism. The energy contribution of WT RAD51 polymerisation enables RAD51 to overcome the entropic penalty to binding flexible DNA (here depicted as ssDNA). F86E RAD51 has a lower RAD51 protomer–protomer affinity and a consequent lower protomer–protomer energy contribution to polymerisation, and thus cannot overcome the entropic penalty for binding to flexible DNA. Both WT and F86E RAD51 can overcome the modest entropic penalty to binding stiff DNA (here depicted as dsDNA). Thick arrows indicate fast capture and fast unbinding, while thin arrows indicate slow capture and slow unbinding.

B   As in Fig 4B, F86E RAD51 was injected at the indicated concentration to measure association and dissociation kinetics. The flexibility of respective DNA, the expected relative entropic penalties upon F86E RAD51 binding to each DNA molecule and protomer–protomer binding energy contribution of F86 RAD51 are indicated in grey, green and blue boxes. H and L shown in each box indicate high and low, respectively.

C   The mean number of F86E RAD51 bound to respective DNA oligonucleotides, as measured by SPR (dotted lines) and corresponding model fits (solid lines) using the model shown in panel (D). Using ABC-SMC, individual forward rates ($k_p$) and reverse rates ($k_{pr}$) were fitted to each experimental curve, while a single $K_D$ was fitted to all experimental curves.

D   Kinetic representation of the ordinary differential equation (ODE) model describing F86E RAD51 polymerisation on ssDNA and dsDNA, consisting of three parameters: $k_p$ (polymerisation forward rate), $k_{pr}$ (polymerisation reverse rate) and $K_D$ (protomer–protomer dissociation constant). The $K_D$ predicts the concentrations of F86E RAD51 polymers of variable length in solution, while $k_p$ and $k_{pr}$ predict the speed of formation of F86E RAD51 polymers on ssDNA and dsDNA. A simplified model compared to the models in Fig 3A and B was used for F86E RAD51 because the kinetics of F86E RAD51 to the dN-5, dN-5p, dN-8, dN-8p, dN-11, dN-11p, dN-14 and dN-17 were not systematically assessed due to low F86E purification yield. The predicted $K_D$ value for F86E is shown. Mean value of mode particles ± 1 SD derived from the ABC-SMC fits ($n = 3$).

E   Bar plots of fitted $k_p$ values for the SPR curves. Mean values of mode particles ± 1 SD derived from the ABC-SMC fits ($n = 3$). Un-paired, one-tailed Mann–Whitney–Wilcoxon tests. *$P \leq 0.05$; ns, non-significant.

F   Predicted concentrations of F86E RAD51 polymers in solution at equilibrium as a function of F86E RAD51 monomer concentration (150 nM, 3 µM, 30 µM), prior to RAD51 injection onto DNA-coated SPR CM5 chips.

G   Predicted % F86E RAD51 within each polymer state in solution at equilibrium as a function of F86E RAD51 monomer concentration (150 nM, 3 µM, 30 µM), prior to RAD51 injection onto DNA-coated SPR CM5 chips. In panels (F, G), the $K_D$ was fixed to the mean value identified in Fig 5D (i.e. $K_D = 10.2$ µM), and the concentration of F86E RAD51 monomer concentration was varied accordingly. In panel (G), the % F86E RAD51 values were calculated by multiplying the polymer concentrations by their respective polymer length and dividing each value by the total RAD51 monomer concentration (i.e. % F86E RAD51 = [$n$-mer] * $n$/[F86E RAD51$_{monomer}$]). [F86E RAD51$_{monomer}$] was calculated via Coomassie staining image quantification using a BSA standard curve.

H   Bar plots of fitted $k_{pr}$ values for the SPR curves. Mean values of mode particles ± 1 SD derived from the ABC-SMC fits ($n = 3$). Un-paired, one-tailed Mann–Whitney–Wilcoxon tests. *$P \leq 0.05$; ns, non-significant.

Source data are available online for this figure.

via a combination of SPR, SAXS and mathematical modelling to understand how RAD51 discriminates between ssDNA and dsDNA. Based on our observations, we propose the Dna molecUlar flExibiliTy (DUET) model, which describes how RAD51 polymerises on ssDNA in preference to dsDNA. RAD51 has a faster polymerisation rate constant on flexible DNA because (i) flexible DNA explores more conformations compatible with RAD51 binding per unit time (the BTC mechanism; Fig 4A), and (ii) the RAD51 protomer–protomer binding energy overcomes the entropic penalty to form RAD51 polymers on flexible DNA (the EPC mechanism; Fig 5A). It is important to highlight, however, that the formation of RAD51 polymer on a separate DNA molecule is a highly complex reaction, and various energies of the system apart from RAD51 and DNA, such as the ions and the water molecules, are expected to influence the enthalpy of the whole system. Regardless, our study indicates that the binding energy between RAD51 protomers has a strong impact in overcoming the entropic penalty in RAD51 binding to flexible DNA, contributing, at least in part, to the enthalpic compensation.

The diploid human genome consists of 6.4 billion base pairs, and ~ 50 endogenous DSBs are estimated to occur in every cell cycle (Vilenchik & Knudson, 2003). In human cells, the resection step of HR-mediated DSB repair can generate ssDNA overhangs up to 3.5 k nucleotides in length (Zhou *et al*, 2014), which then serve as a platform for RAD51 polymerisation. Given that ssDNA overhangs would constitute only ~ 0.005% of the total genomic DNA (3.5 k nucleotides × two overhangs × 50 DSBs/6.4 billion base pairs), the polymerisation of RAD51 on resected ssDNA needs to be greatly directed. The average nuclear concentration of RAD51 is estimated at ~ 100 nM (Reuter *et al*, 2014), and several RAD51 mediators,

such as BRCA2, PALB2 and RAD52, contribute to increase the RAD51 local concentration at DSBs and/or RAD51 binding to ssDNA, while preventing its association with dsDNA (Miyazaki *et al*, 2004; Buisson *et al*, 2010; Jensen *et al*, 2010; Carreira & Kowalczykowski, 2011; Zhao *et al*, 2015; Ma *et al*, 2017a). Nonetheless, to date, *in vivo* evidence that these RAD51 mediators are sufficient to promote RAD51 polymerisation on DSB-derived ssDNA in preference to the bulk of undamaged dsDNA is limited.

Our approach established in this study, combining SPR and mathematical modelling, proved to be a powerful way to uncover previously poorly understood properties of RAD51. This pipeline has unique advantages over biophysical studies exploiting optical tweezers (Van der Heijden *et al*, 2007; Candelli *et al*, 2014; Brouwer *et al*, 2018), where RAD51 binding to a long, typically 10 k nucleotides DNA has been evaluated via the force and length alteration of tethered DNA. In contrast, SPR offered real-time, sensitive and label-free detection of RAD51 assembly and disassembly on a series of short DNAs. The probability of multi-nucleation of RAD51 on a single DNA molecule was thus constrained, minimising the complexity of experimental errors used in our model fitting, in which RAD51 polymers both in solution and on DNA at defined concentrations were also taken into account. These analyses collectively allowed us to conclude that differences in RAD51-DNA binding kinetics are associated with DNA flexibility and that RAD51 preferentially polymerises on flexible DNA. It should be pointed out, however, that the current study is built on RAD51 kinetics in the presence of Ca$^{2+}$, which blocks RAD51-mediated ATP hydrolysis. This condition slows down RAD51 dissociation from DNA, enabling highly sensitive detection of RAD51 binding to short DNA substrates. We envision that future studies using this pipeline under

conditions allowing RAD51 ATP hydrolysis and/or in the presence of other HR mediator proteins, such as BRCA2 and RAD51 paralogs (Shahid *et al*, 2014; Taylor *et al*, 2016), will shed further light on the dynamics of RAD51 polymerisation on DNA.

It is important to note that, under physiological conditions, resected ssDNA is first bound by RPA prior to RAD51 polymerisation. RPA is a heterotrimer complex with six OB folds, four of which can associate tightly with ssDNA in a stepwise manner (Zou *et al*, 2006; Fan & Pavletich, 2012). RPA binding to ssDNA is believed to eliminate ssDNA secondary structures to facilitate RAD51 polymerisation. Significantly, previous kinetic studies have demonstrated a rapid exchange of ssDNA-bound RPA with free RPA available in solution, a phenomenon defined as "facilitated exchange" (Gibb *et al*, 2014; Ma *et al*, 2017b). It has further been proposed that microscopic dissociation of the individual DNA-binding domains of RPA may provide a landing site for downstream players such as RAD51 (Chen & Wold, 2014; Chen *et al*, 2016). While one RAD51 protomer binds to three nucleotides of ssDNA (Short *et al*, 2016), one RPA heterotrimer has a footprint of 30 nucleotides, which is enough to accommodate up to ten RAD51 molecules. Interestingly, the DNA footprint of a RAD51 pentamer nucleus corresponds to about half of the RPA DNA footprint. Formation of a stable nucleus of five RAD51 molecules on ssDNA may take place when RPA is macroscopically bound but two of its most terminal DNA-binding domains are microscopically dissociated. This situation agrees with the kinetics of RPA domain conformational dynamics (Pokhrel *et al*, 2019). Hence, it is reasonable to speculate that microscopic dissociation of RPA is enough to provide highly flexible ssDNA on which RAD51 can nucleate and elongate.

Another major unanswered question regarding RAD51 polymerisation is how RAD51 can distinguish ssDNA generated by DSB resection from ssDNA formed during normal cellular processes, such as DNA replication and transcription. Spontaneous RAD51 polymerisation on all ssDNA would be problematic, as it may trigger undesired, toxic recombination events or disruption of DNA replication and transcription, causing genomic instability. In this context, the DUET model is particularly appealing: ssDNA generated during transcription and replication does not have free ends and could therefore be less flexible compared to resected ssDNA with a free 3′ overhang. This in turn may limit RAD51 polymerisation on transcription- and replication-derived ssDNA, while promoting it on flexible ssDNA at DSBs. Intriguingly, a recent report demonstrated that poly(dA) stretches at replication forks are more vulnerable to DNA damage as they are unprotected by RPA, forming early-replicating fragile sites (Tubbs *et al*, 2018). Our study further suggests that such poly(dA)-associated DNA damage is less efficiently repaired by RAD51-mediated HR, increasing genome instability at these loci.

Beyond HR repair, this study suggests a general framework to understand how DNA-binding proteins are recruited to sites of DNA damage. Numerous DNA repair machineries are composed of multiple subunits with several binding interfaces to DNA and other cofactors. It is therefore tempting to speculate that broken DNA ends may enhance recruitment of such repair complexes simply due to increased DNA flexibility. In line with this notion, increased mobility of broken DNA has been demonstrated in mammalian cells (Aten *et al*, 2004; Cho *et al*, 2014; Aymard *et al*, 2017). Hence, this work

presents a conceptual advancement in linking DNA repair and DNA flexibility, adding an important dimension which should be taken into account when assessing repair processes both in biochemical assays and in cellular contexts.

## Materials and Methods

### RAD51 mutagenesis

The bacterial expression vector pET11d (Merck-Millipore) carrying the human WT RAD51 (WT-pET11d) was used as a template for the PCR-mediated QuikChange site-directed mutagenesis (Agilent Technologies) to introduce F86E substitution (F86E-pET11d) with a forward primer 5′-GCTAAATTAGTTCCAATGGGTGAGACC ACTGC AACTGAATTCCACC–3′ and a reverse primer 5′-GGTGGAATTCA GTTGCAGTGGTCT CACCCATTGGAACTAATTTAGC–3′.

### RAD51 protein purification

To express RAD51, Rosetta 2 (DE3) pLysS cells (Novagen) carrying the WT-pET11d or F86E-pET11d were grown in LB media containing 100 μg/ml ampicillin and 25 μg/ml chloramphenicol, and by adding 0.5 mM IPTG at $OD_{595} = 0.6$, protein expression was induced. Cell pellets were resuspended in PBS and mixed with the equal volume of lysis buffer (3 M NaCl, 100 mM Tris–HCl pH 7.5, 4 mM EDTA pH 8, 20 mM ß-mercaptoethanol, Sigma Protease Inhibitor Cocktail [Sigma]). The suspension was sonicated and spun at 20k rpm with a 45 Ti rotor (Beckman). The supernatant was slowly mixed with 0.1% polyethylenimine at 4°C for 1 h and spun at 20k rpm using a 45 Ti rotor to remove DNA. The supernatant was then slowly mixed with an equal volume of 4 M ammonium sulphate (2 M final concentration) at 4°C for 1 h and spun at 10k rpm using a JA-17 rotor (Beckman). Pellets containing RAD51 were suspended in 25 ml of resuspension buffer (0.5 M KCl, 50 mM Tris–HCl pH 7.5, 1 mM EDTA, 2 mM DTT, 10% glycerol) and spun again at 20k rpm to remove residual DNA. The supernatant was then dialysed overnight at 4°C in TEG buffer (50 mM Tris pH 7.5, 1 mM EDTA, 2 mM DTT, 10% glycerol) containing 200 mM KCl (TEG200). For F86E RAD51, the cell lysate was prepared as for WT RAD51 and dialysed overnight at 4°C in TEG buffer containing 50 mM KCl (TEG75).

RAD51 purification was carried out by chromatography at 4°C using the AKTA Pure Protein Purification System (GE Healthcare). For WT RAD51, the dialysed WT RAD51 containing sample was loaded onto a 5 ml HiTrap Heparin column (GE Healthcare) and eluted via a linear gradient of 200–600 mM KCl. The peak fractions were pooled and dialysed in TEG200, and concentrated on a 1 ml HiTrap Q column (GE Healthcare) followed by the isocratic elusion with TEG buffer containing 600 mM KCl (TEG600). Peak fractions containing WT RAD51 were dialysed in SPR buffer (150 mM KCl, 20 mM Hepes pH 7.5, 2 mM DTT, 10% glycerol), aliquoted, snap frozen and stored at −80°C. Similarly, the dialysed F86E RAD51 containing cell lysate was fractionated through a 5 ml HiTrap Heparin column, but with a linear gradient of 75–600 mM KCl. The flow-through and F86E RAD51 peak fractions were pooled and dialysed in TEG buffer containing 50 mM KCl (TEG50) and reloaded onto a 5 ml HiTrap Heparin column. Following a 50–600 mM KCl

linear gradient elution, F86E RAD51 peak fractions were pooled and dialysed in TEG buffer containing 100 mM KCl (TEG100). The sample was concentrated on a 1 ml HiTrap Q column followed by TEG600 isocratic elution. Peak fractions were applied on a 24 ml Superdex200 10/300 GL size-exclusion column (GE Healthcare), and fractions containing monomeric F86E RAD51 were pooled and dialysed in TEG100. F86E RAD51 was then re-concentrated using a 1 ml HiTrap Q column and TEG600 isocratic elution. The peak fractions were dialysed in SPR buffer, aliquoted, snap frozen and stored at −80°C.

## Multi-angle light scattering

To confirm the monomeric status of F86E RAD51, the peak size-exclusion chromatography F86E RAD51 elution fraction was serially diluted (5, 1:1 serial dilutions) and loaded onto a Superdex200 10/300 GL size-exclusion column (GE Healthcare) equilibrated with MALS Buffer (150 mM KCl, 50 mM Tris pH 7.5, 1 mM EDTA, 2 mM DTT). Each elution was analysed using a Wyatt Heleos8 + 8-angle light scatterer linked to a Shimadzu HPLC system comprising LC-20AD pump, SIL-20A Autosampler and SPD20A UV/Vis detector. Data collection was carried out at the Department of Biochemistry, University of Oxford. Data analysis was carried out using the ASTRA software (Wyatt).

## DNA oligonucleotides and duplex synthesis

The dN-5p, dN-8p and dN-11p dsDNA sequences were designed as two complimentary ssDNA sequences connected via two units of hexaethylene glycol (flexible linker) (Table 2). The dN-5, dN-8 and dN-11 ssDNA molecules were designed using one of the two corresponding dsDNA annealing sequences. The dN-14 and dN-17 ssDNA molecules were designed by extending one of the two dN-11p dsDNA annealing sequences. Standard DNA phosphoramidites, solid supports, 3′-Biotin-TEG CPG and additional reagents were purchased from Link Technologies Ltd and Applied Biosystems Ltd. All oligonucleotides were synthesised on an Applied Biosystems 394 automated DNA/RNA synthesiser using a standard 1.0 μmole phosphoramidite cycle of acid-catalysed detritylation, coupling, capping and iodine oxidation. Stepwise coupling efficiencies and overall yields were determined by the automated trityl cation conductivity monitoring facility and in all cases were > 98.0%. All β-cyanoethyl phosphoramidite monomers were dissolved in anhydrous acetonitrile to a concentration of 0.1 M immediately prior to use. The coupling time for normal A, G, C and T monomers was 60 s, and the coupling time for the hexaethylene glycol phosphoramidite monomer (from link) was extended to 600 s. Cleavage of the oligonucleotides from the solid support and deprotection was achieved by exposure to concentrated aqueous ammonia solution for 60 min at room temperature followed by heating in a sealed tube for 5 h at 55°C. Purification of oligonucleotides was carried out by reversed-phase HPLC on a Gilson system using a Brownlee Aquapore column (C8, 8 mm × 250 mm, 300 Å pore) with a gradient of acetonitrile in triethylammonium bicarbonate (TEAB) increasing from 0% to 50% buffer B over 20 min with a flow rate of 4 ml/min (buffer A: 0.1 M triethylammonium bicarbonate, pH 7.0, buffer B: 0.1 M triethylammonium bicarbonate, pH 7.0 with 50% acetonitrile).

Elution of oligonucleotides was monitored by ultraviolet absorption at 295 or 300 nm. After HPLC purification, oligonucleotides were freeze-dried and then dissolved in water without the need for desalting. All oligonucleotides were characterised by negative-mode HPLC–mass spectrometry using either a Bruker micrO-TOFTM II focus ESI-TOF mass spectrometer with an Acquity UPLC system, equipped with a BEH C18 column (Waters), or a Waters Xevo G2-XS QT mass spectrometer with an Acquity UPLC system, equipped with an Acquity UPLC oligonucleotide BEH C18 column (particle size: 1.7 μm; pore size: 130 Å; column dimensions: 2.1 × 50 mm). Data were analysed using Waters MassLynx software or Waters UNIFI Scientific Information System software.

## Small-angle X-ray scattering

The flexibilities of the ssDNA dT-50, ssDNA dN-50, ssDNA dA-50 and the dsDNA dN-50p molecules were assessed using SAXS at the Diamond Light Source (Harwell, UK). SAXS data were collected using a size-exclusion KW 402.5 (2.4 ml) column (Shodex). Fifty-microliter of DNA sample was injected, and elution was carried out at 37°C at 75 μl/min using SPR running buffer in the absence of glycerol and BSA. The flexible cylinder model was fit to the four scattering datasets within the 0.0037–0.27 [1/Å] range to derive the persistence lengths. Data plotting and fitting were carried out using the SasView software for SAXS data analysis.

## Surface plasmon resonance

The binding kinetics of WT and F86E RAD51 on ssDNA and dsDNA were assessed via SPR using a Biacore T200 (GE Healthcare). CM5 SPR chip flow cells were activated by injecting 100 μl of a 1:1 *N*-Hydroxysuccinimide (NHS) and ethyl(dimethylaminopropyl) carbodiimide (EDC) mix at 10 μl/min. One hundred microliter of purified streptavidin (1 mg/ml) was then injected over the flow cells at 10 μl/min. Unbound amine groups were then deactivated by injecting 50 μl of 1 M ethanolamine-HCl. Fifty microliter of 10 mM glycine pH 2.5 was injected to remove uncoupled, sterically bound streptavidin. For all experiments, ~ 8–15 RU of biotinylated DNA substrates were immobilised onto either flow cells 2, 3 or 4. After DNA immobilisation, 10 μl of purified biotin was injected over all four flow cells at 10 μl/min to block the remaining free streptavidin binding sites. Chip activation and ligand immobilisation steps were carried out at 25°C. Using high-performance injections, the CM5 chip surfaces were then primed via 10 injections of 10 μl SPR running buffer (150 mM KCl, 20 mM Hepes pH 7.5, 2 mM DTT, 5 mg/ml BSA, 2.5 mM ATP pH 7.5, 10 mM $CaCl_2$, 10% glycerol) at 30 μl/min. WT or F86E RAD51 diluted in SPR running buffer to the specified concentration was injected at 30 μl/min (association), followed by the injection of SPR running buffer for 20 min at 30 μl/min (dissociation) at 37°C.

## SPR data processing

Surface plasmon resonance traces were processed using the BIAevaluation software (GE Healthcare) as following. First, the negative control flow cell trace (Fc1 or Fc3) was subtracted from the experimental flow cell trace. The trace was then vertically and horizontally aligned, such that the start of the protein injection

occurs at time = 0 s, response = 0 RU. Subsequently, the drift trace between the negative control flow cell and the experimental flow cell during the final priming injection was subtracted from the experimental flow cell trace (double-referencing) (Myszka, 1999). All experimental curves were normalised using the following equation:

$$N = S/(L * (M_{RAD51}/M_{DNA}))$$

where $N$ is the normalised signal (mean number of RAD51 molecules per DNA oligonucleotide), $S$ is the signal in RU units (1 RU ~ 50 pg/mm$^2$), $L$ is the amount of DNA ligand immobilised onto the experimental flow cell (RU), $M_{RAD51}$ is the molecular weight (kDa) of RAD51 (~ 37 kDa) and $M_{DNA}$ is the molecular weight (kDa) of the immobilised DNA molecule. The normalised signal represents the mean number of RAD51 molecules bound to an immobilised DNA molecule and has a predicted maximum at 16 (i.e. the predicted maximum number of RAD51 molecules bound to a ssDNA or dsDNA 50mer).

### Mathematical model development and analysis

The ODE model describing RAD51 polymerisation on DNA consists of an equilibrium sub-model and a kinetic sub-model. The equilibrium sub-model was formulated using second-order mass action kinetics via the rule-based modelling language BioNetGen (Harris *et al*, 2016) and consists of 16 non-linear ODEs describing the formation of RAD51 polymers in solution up to a maximum length of 16 (i.e. the predicted maximum number of RAD51 molecules binding to a DNA 50-mer (Short *et al*, 2016)). Each ODE describes the rate of change in concentration of a RAD51 polymer in solution with respect to time. They are non-linear because monomeric RAD51 is consumed in the process of polymerisation to form RAD51 polymers. In this sub-model, we assume any RAD51 $n$-mer ($1 \leq n < 16$) can bind to any other RAD51 $m$-mer ($1 \leq m < 16$) to form a RAD51 ($n + m$)-mer ($1 < n + m \leq 16$). In addition, any RAD51 $k$-mer can fall apart in every possible combination of $m$-mers and $n$-mers ($k = n + m$; e.g. a pentamer can fall apart to form a monomer and a tetramer, or a dimer and a trimer). The single dissociation constant ($K_D$) describing all of these pairwise interactions is a fit parameter derived from the SPR data. The model is evaluated at equilibrium to calculate the concentration of each RAD51 polymer in solution, prior to the SPR injection. The concentration distribution of RAD51 polymers in solution depends on the fit parameter $K_D$. The predicted concentrations of RAD51 polymers are then inserted into the kinetic sub-model.

The kinetic sub-models describe the formation of RAD51 polymers on DNA. Importantly, during SPR injections the concentrations of RAD51 polymers in solution in the flow cell remain constant because RAD51 is continuously replenished by flow. For this reason, the kinetic sub-models were formulated using first-order mass action kinetics via the rule-based modelling language BioNetGen (Harris *et al*, 2016) and each consists of 17 linear ODEs describing the change in concentration of RAD51-polymer-bound DNA molecules. The extra ODE describes unbound DNA molecules. See Supplementary Information for full details of model development.

For WT RAD51, ABC-SMC (Toni *et al*, 2009) was carried in MATLAB 2016b to determine a unique set of parameters for the

equilibrium and kinetic sub-models that can simultaneously explain the ssDNA dN-X data (Figs 2B, 3A and EV1) and the dsDNA dN-Xp data (Figs 2D, 3B and EV1). dN-X & dN-Xp $K_D$, dN-X $k_p$, $k_q$, $k_s$, and dN-Xp $k_p$ are well determined. dN-X $k_u$ and dN-Xp $k_u$ were undetermined, due to the fact that (i) no RAD51 binding was observed to the dN-8 and dN-5p oligonucleotides, and (ii) the concentration of RAD51 monomers, dimers and trimers in solution at 3 µM [WT RAD51] is low relative to longer polymers (Fig 3C). Finally, we could only determine a well-defined upper bound for the dsDNA stable reverse rate constant (dN-Xp $k_s$ < 10$^{-3}$). For F86E RAD51, ABC-SMC was carried out in MATLAB 2016b to determine a unique set of parameters for the equilibrium sub-model and a simplified version of the kinetic sub-model (Fig 5D) that can simultaneously explain the ssDNA dT-50, dN-50, dA-50 and the dsDNA dN-50p data (Figs 5C,E,H, and EV5). All parameters except dN-50p $k_{pr}$ were well determined. dN-50p $k_{pr}$ has a well-defined upper bound (i.e. dN-50p $k_{pr}$ < 10$^{-3}$).

Additional details can be found in the Appendix Methods and in the Source Data files.

### Statistical analysis

Experiments were carried out in triplicates ($n = 3$) and, for this reason, the non-parametric Mann–Whitney–Wilcoxon test was used to test for significant differences between the different fitted RAD51 polymerisation rates and lifetimes (Figs 4D and E, and 5E and H). All tests were carried out using GraphPad Prism 7.

# Data availability

All SPR datasets produced in this study are available via open science framework: https://osf.io/28cqv/.

**Expanded View** for this article is available online.

### Acknowledgements

FE and OD are supported by Wellcome Trust Senior Research Fellowships in Basic Biomedical Science (101009/Z/13/Z and 207537/Z/17/Z, respectively). JA is supported by National Science Foundation grant (DMS 1454739). TB is thankful for the support by Biotechnology and Biological Sciences Research Council (BB/J001694/2). FP is a recipient of the Systems Biology Doctoral Training Centre Scholarship, funded by the Engineering and Physical Sciences Research Council. We thank Marcus Bridge for assistance with SPR, Nicola Trendel for assistance with ABC-SMC, David Staunton for the SEC-MALS analysis of the RAD51 mutant and Robert Rambo for the DNA oligonucleotides SAXS analysis. We also acknowledge the use of the University of Oxford Advanced Research Computing (ARC) facility in carrying out this work (https://doi.org/10.5281/zenodo.22558).

### Author contributions

FE, OD and FP conceived and planned the project. FP conducted all SPR experiments, protein purification and mathematical modelling. AES and TB designed and generated the short DNA oligonucleotides for SPR experiments. JA contributed to the conceptualisation of the thermodynamic impact of polymerisation. FE, OD and FP wrote the manuscript with input from all contributing authors.

### Conflict of interest

The authors declare that they have no conflict of interest.

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
