## [Review Process File · The EMBO Journal]

Molecular flexibility of DNA as a key determinant of RAD51 recruitment

Federico Paoletti, Afaf El-Sagheer, Jun Allard, Tom Brown, Omer Dushek and Fumiko Esashi.

Review timeline:

Submission date:	19 th July 2019
Editorial Decision:	15 th August 2019
Revision received:	21 st November 2019
2nd Editorial decision:	11 th December 2019
Revision received:	16 th December 2019
Accepted:	16 th December 2019

Editor: Hartmut Vodermaier

Transaction Report:

1st Editorial Decision

15th August 2019

Thank you for submitting your manuscript on molecular flexibility as determinant of RAD51 recruitment to The EMBO Journal. We have now received comments from three expert referees, copied below for your information. As you will see, the referees appreciate the conceptual rationale for the study and also consider the conclusions involving entropy reduction compensated by an enthalpic gain potentially quite interesting. At the same time, all referees remain still unconvinced that the present dataset offers sufficiently definitive and unambiguous support for these conclusions. A key issue in this regard is that the invoked enthalpic compensation would need to be directly demonstrated by isothermal titration calorimetry or other possible complementary methods (see referees 1 & 2). Another key concern is that the causal role of DNA flexibility would require stronger evidence from independent validation with alternative methods, such as molecular tweezers (see referees 2 and 3).

In light of these comments, I feel that publication of the paper in The EMBO Journal would still be somewhat premature at the present stage. However, should you be able to extend the study beyond the present approaches and obtain decisive complementary support for the main conclusions, then we would be happy to consider a revised manuscript further.

REFeree REPORTS

Referee #1:

The biological importance of this work comes from the question the authors pose, namely, how recombinases overcome the entropy associated with straightening of the ssDNA upon formation of the nucleoprotein filament. I don't think I saw this point ever being discussed before, but its importance is quite evident since our current structural knowledge of the RAD51 (and related recombinases) filament formation suggests that the interactions between the recombinase and the DNA should be pretty similar for ssDNA vs. dsDNA. Recombinase (more so RecA, but RAD51 as well) display preference for ssDNA despite higher entropic cost of strengthening ssDNA.

To address the mechanism of the nucleoprotein filament nucleation by RAD51 recombinase the authors employed SPR (a label-free approach to analyze the kinetics of macromolecular interactions) and kinetic modeling. The numerical modeling, the authors developed, is a definite value added part of the work. This is especially true for modeling SPR data for an oligomeric protein such as RAD51. By combining SPR and modeling the authors observed:

- 1) Faster rate of nucleoprotein filament formation on ssDNA, but similar stability of the formed filament on ssDNA and dsDNA 50-mer.
- 2) A requirement of at least 5 bound RAD51 monomers to form a stable nucleus. These data agree nicely with observations by Candelli et al 2014 and Subramanyam et al 2016, but also provide a more precise kinetic rate constants, which is a new and important information. Note that the latter reference is on the reference list, but does not seem to be mentioned in the body of the text.
- 3) That on dsDNA, 2-3 Rad51 molecules are sufficient for form a stable nucleus.
- 4) That RAD51 polymerases faster on more flexible ssDNA, which the authors propose are due to a significant enthalpic contribution of the protomer-protomer interaction. To confirm this, the authors analyzed the RAD51 F86E mutant, which has reduced RAD51-RAD51 interaction and is a monomer in solution.

Overall, the experiments and the modeling are expertly performed and the idea the authors explore in this manuscript is truly original. I have only one concern:

In their model, the authors suggest that the reason underlying the specificity is enthalpic compensation for the entropy cost. Which is possible, but needs a direct prove. They only consider the entropic cost of constraining the DNA molecule, but there could be other contributors, such as solution (water and ions) and possible changes in RAD51 flexibility. I see two ways how the authors can quantify/compare the enthalpic component:

- 1) ITC (this may require establishing a troubleshooting the experimental scheme), or
- 2) They can do the van't Hoff analysis using the SPR system they already established. A two-three additional temperatures for WT and mutant nucleoprotein filament assembly on one of the substrates (4 - 6 additional binding experiments) should be sufficient to test whether the enthalpy is indeed the main contributor.

Minor points:

Lines 145-146: should it be "protomer" instead of "promoter"?

Discussion (lines 300 and below) talks about facilitated exchange of RPA referencing Ma et al 2017 paper (Greene lab). A more appropriate paper from the same lab, which suggested the mechanism for such an exchange is Gibb et al 2014. Several labs, including Greene and Wold lab (Chen et al 2016 NAR, reviewed in Chen and Wold 2014 Bioessays) proposed that microscopic dissociation of the individual DNA-binding domains of RPA may provide a landing site for the downstream players, such as RAD51. The 5-RAD51 stable nucleus would correspond to about a half of the RPA-occluded binding site and may take place when RPA is macroscopically bound, but two of its most terminal DBDs (e.g. C and D) and microscopically dissociated. This situation agrees very well also with the kinetics of RPA domain conformational dynamics directly observed by Pokhrel et al 2019 NSMB and with the effect of yeast Rad52 protein on this dynamics (same paper).

Referee #2:

EMBOJ-2019-103002, corr. author Dr. Esashi

"Molecular flexibility of DNA as a key determinant of RAD51 recruitment"

The paper by Paoletti et al. describes biochemical analysis on the DNA binding of human RAD51 protein involved in homologous recombination using surface plasmon resonance (SPR). The study particularly focuses on the effect of DNA flexibility (as well as single-stranded versus double-stranded) and of DNA length on association and dissociation rates of RAD51 to the DNA as well as polymerization rate on the DNA. The authors found that the flexibility of the DNA is a critical determinant of RAD51 polymerization on the DNA. The study also addressed the question how "rigid" RAD51 filaments are formed on "flexible" ssDNA by overcoming an entropic penalty and proposed enthalpic contribution by RAD51 polymerization would compensate the penalty. The works in the paper were carried out in a good care, the results are in a good quality and main conclusions are supported by the data. However, there seem more to support their conclusion in terms of biological significance of findings in the paper on RAD51-assembly in a cell for homology search and strand exchange. And also the authors need more words to readers in the journal, who are

not familiar to this kind of biochemistry and modeling.

Major points:

1. Correlation between DNA flexibility (Table 1) and RAD51 polymerization rate (Fig. 3D) does not look strong. Particularly dA50 and dN-50p with different persistent length (4.7 and 7.3, respectively) show the similar polymerization rate (0.0005 and 0.0004, respectively). Thus, the polymerization does not seem to explain the different binding affinities of RAD51 to dA50 and dN-50p. More interesting, rather than wild type, RAD51-F86E mutant protein shows correlation between persistent length and polymerization rates (Fig. 4E), which suggests that rather than WT RAD51, RAD51-F86E seems to sense flexibility of DNAs.
2. In the same line to #1 comment, as discussed by the authors (page 13, line 316-320), it would be great if the authors can perform SPR binding analysis of RAD51 protein using dN-50p with "double" tethering to the chip, which reduces the flexibility (entropy).
3. The authors proposed a bend-to-capture (BTC) mechanism to explain how the DNA flexibility affects polymerization. However, as authors pointed out in Figure 3A, the 2nd capture requires the dissociation of RAD51 from the 1st binding site. However, in their experiments, the authors used a condition in which RAD51 shows little dissociation from the DNA in the presence of Ca²⁺ (very long lifetime shown in Fig. 3E). How could very slow dissociation promote BTC? Thus, BTC does not seem to explain faster polymerization of RAD51 on more flexible DNAs. Or need more explanation in the text, if I am missing some points.
4. The authors proposed that enthalpic contribution of RAD51 polymerization is a key determinant to overcome entropic penalty on the flexible DNA. Is there an experimental evidence that RAD51 polymerization follows exothermic reaction by using isothermal titration calorimetry (ITC)?
5. In Summary (page 2, line 34) and other parts (page 4, line 85), the authors used "mechano-sensor" for RAD51. What does this mean? The authors need more explanation on the biological significance of mechano-sensor in homologous recombination. Since, as noted by the authors, ssDNAs produced in vivo are much longer than substrate used here and thus, intrinsically flexible or once coated with RPA, becomes more rigid. Without the additional information, it is premature to conclude that RAD51 is a mechano-sensor.

Minor points:

1. For DNA binding analysis by SPR, it would be nice to show the concentration effect on the binding curves to different types of DNAs. And also it would be nice to explain what Y-axis means. For example, in Fig. S1B, there is difference in SPR signal (shown in Y-axis) between ssDNA and dsDNA in the same length.
2. One caveat in the modeling of Figure 2A and B etc, the authors do not seem assume that a single RAD51 multimer (Page 8, line 178-179); e.g. trimer binds to 5mer, which has a single RAD51 binding site (once RAD51 trimer binds to the 5mer, this triggers the dissociation of two RAD51?). Or this product might not show any dynamics. Consideration of these reactions should be important, since, under their condition, the authors indeed showed that RAD51 forms relatively long polymers whose average more than 10mer (site size should be 30 nucleotides) as shown in Fig. 2C and D.
3. Page 5, line 101: WT RAD51 distinguishes ssDNA from dsDNA through faster polymerization on ssDNA. What data support this? Could be slop in the association phase. Need more word to readers.
4. Page 6, line 134: What is "ODE"?
5. Page 8, line 184: 2.6+/-1.8 should be 2.6+/-"0.18".
6. Page 11, line 254: Add (Fig. 4C) after dN-50p.

Referee #3:

The RAD51 proteins binds both single and double-stranded DNA. Therefore, it has been of considerable interest to understand mechanisms that allow RAD51 to preferentially bind to ssDNA, as only this binding permits homologous recombination.

Paoletti et al here base their experiments on previous findings that indicated that RAD51 polymerizes faster on single as opposed to double stranded DNA. To explain how RAD51 can discriminate between the two, they use a combination of SPR and mathematical modeling to suggest that flexibility of DNA plays an important role. In particular, they use various DNA sequences (dT,

dA and scrambled sequence) that differ in terms of rigidity and find a correlation between polymerization rate and the flexibility/rigidity of the DNA sequences used. Likewise, dissociation of RAD51 from DNA is dependent on DNA rigidity. This is suppressed when using wt RAD51 mutant that stabilizes DNA, but is revealed when using a mutant with impaired RAD51-RAD51 interaction that renders the nucleoprotein filament less rigid.

Generally, the manuscript is well written and the experiments are well-explained. My main concern is whether the observations are correlative or causative. In other words, what if RAD51 prefers to bind dT, and it has little to do with DNA flexibility. In this regard, the used method has a limitations. It would be very nice to complement the results with an alternative method. I.e. molecular tweezers have been used previously to monitor assembly of RAD51 filaments, and the method seems to be much better suited to verify author's conclusions, as variation of applied force would directly affect DNA flexibility, without concern about DNA sequence difference.

My other comments are discussion points:

- The authors refer to the studies from the Rothstein and Gasser labs, which observed increased DNA mobility upon DNA breakage. This increased mobility was, however, clearly dependent on Rad51 and checkpoint, and it therefore cannot explain or facilitate RAD51 recruitment. The way the authors refer to these studies is somewhat misleading.
- The authors propose that displacement of RPA will yield ssDNA of 30 nt in length, which might yield highly flexible DNA. It seems unlikely to me that the exposed DNA would be highly flexible, it if is surrounded on both ends with RPA-coated ssDNA of hundreds of nucleotides in length.

1st Revision - authors' response

21st November 2019

Response to referees' comments

We would like to thank the referees for their careful consideration of our manuscript and providing valuable suggestions. Below, we provide a point-by-point response, outlining how we have addressed and/or clarified the issues raised by the referees.

Referee #1:

The biological importance of this work comes from the question the authors pose, namely, how recombinases overcome the entropy associated with straightening of the ssDNA upon formation of the nucleoprotein filament. I don't think I saw this point ever being discussed before, but its importance is quite evident since our current structural knowledge of the RAD51 (and related recombinases) filament formation suggests that the interactions between the recombinase and the DNA should be pretty similar for ssDNA vs. dsDNA. Recombinase (more so RecA, but RAD51 as well) display preference for ssDNA despite higher entropic cost of strengthening ssDNA.

To address the mechanism of the nucleoprotein filament nucleation by RAD51 recombinase the authors employed SPR (a label-free approach to analyze the kinetics of macromolecular interactions) and kinetic modeling. The numerical modeling, the authors developed, is a definite value added part of the work. This is especially true for modeling SPR data for an oligomeric protein such as RAD51. By combining SPR and modeling the authors observed:

1) Faster rate of nucleoprotein filament formation on ssDNA, but similar stability of the formed filament on ssDNA and dsDNA 50-mer.

2) A requirement of at least 5 bound RAD51 monomers to form a stable nucleus. These data agree nicely with observations by Candelli et al 2014 and Subramanyam et al 2016, but also provide a more precise kinetic rate constants, which is a new and important information. Note that the latter reference is on the reference list, but does not seem to be mentioned in the body of the text.

3) That on dsDNA, 2-3 Rad51 molecules are sufficient to form a stable nucleus.

4) That RAD51 polymerizes faster on more flexible ssDNA, which the authors propose are due to a significant enthalpic contribution of the protomer-protomer interaction. To confirm this, the authors analyzed the RAD51 F86E mutant, which has reduced RAD51-RAD51 interaction and is a monomer in solution. Overall, the experiments and the modeling are expertly performed and the idea the authors explore in this manuscript is truly original.

We are grateful for the reviewer's recognition of the biological and biophysical importance of this study, as well as the significance of the methodological approach developed to tackle this important question.

I have only one concern:

In their model, the authors suggest that the reason underlying the specificity is enthalpic compensation for the entropy cost. Which is possible, but needs a direct prove. They only consider the entropic cost of constraining the DNA molecule, but there could be other contributors, such as solution (water and ions) and possible changes in RAD51 flexibility. I see two ways how the authors can quantify/compare the enthalpic component: 1) ITC (this may require establishing a troubleshooting the experimental scheme), or 2) They can do the van't Hoff analysis using the SPR system they already established. A two-three additional temperatures for WT and mutant nucleoprotein filament assembly on one of the substrates (4 - 6 additional binding experiments) should be sufficient to test whether the enthalpy is indeed the main contributor.

We appreciate this reviewer's concern. To make interpretation straightforward, we used DNA as the only varying factor in all our SPR analyses while keeping constant other conditions (e.g. ions and temperature: 37°C, which is physiological for the human protein). While different concentrations of RAD51 (WT or mutant) were used, the potential change in RAD51 multimerisation in solution was taken into account during mathematical modelling. In other words, the observed difference can only be attributed to the property of DNA.

We foresee experimental quantification of the enthalpic component of the specific chemical reaction, either by ITC or by the van't Hoff analysis, of polymer (RAD51 filament) formation on a second polymer (DNA) being extremely challenging. As this reviewer recognises, the ITC approach would require significant optimization with little promise; human RAD51, unlike bacterial RecA, is prone to aggregate at high concentrations, so ITC is expected to generate high background noise. We also foresee that conducting SPR at different temperatures, for the van't Hoff analysis, would affect multiple factors, i.e., a single RAD51 molecule, water, ions, as well as long molecules of DNA and mixtures of various RAD51-DNA complexes, making the interpretation extremely complex. To our knowledge, van't Hoff analysis has been conducted to assess enthalpy of one-to-one chemical reactions, but not for highly complex polymer-polymer interactions.

To avoid confusion, we have made clear which biophysical aspects of RAD51/DNA interaction contribute to entropy and enthalpic gain in this revised manuscript: the entropy refers to the intrinsic energy of DNA molecules which depends on DNA persistence length, while the enthalpic gain is the energy incurred by the binding of RAD51 to an existing RAD51 filament, which is independent of DNA persistence length. As the reviewer highlights, there are many other processes that contribute to the overall entropy and enthalpy of the reaction under study, however, we are highlighting the components that we have directly perturbed (DNA flexibility and RAD51-RAD51 binding interface). We hope that this will clarify that our model

invokes these specific changes without claims about the overall reaction entropy and enthalpy.

Minor points:

Lines 145-146: should it be "protomer" instead of "promoter"?

We thank this reviewer for identifying the error. This is now corrected.

Discussion (lines 300 and below) talks about facilitated exchange of RPA referencing Ma et al 2017 paper (Greene lab). A more appropriate paper from the same lab, which suggested the mechanism for such an exchange is Gibb et al 2014. Several labs, including Greene and Wold lab (Chen et al 2016 NAR, reviewed in Chen and Wold 2014 Bioessays) proposed that microscopic dissociation of the individual DNA-binding domains of RPA may provide a landing site for the downstream players, such as RAD51. The 5-RAD51 stable nucleus would correspond to about a half of the RPA-occluded binding site and may take place when RPA is macroscopically bound, but two of its most terminal DBDs (e.g. C and D) and microscopically dissociated. This situation agrees very well also with the kinetics of RPA domain conformational dynamics directly observed by Pokhrel et al 2019 NSMB and with the effect of yeast Rad52 protein on this dynamics (same paper).

We are grateful for this reviewer's excellent input. We have incorporated these points in our revised manuscript.

Referee #2:

EMBOJ-2019-103002, corr. author Dr. Esashi "Molecular flexibility of DNA as a key determinant of RAD51 recruitment"

The paper by Paoletti et al. describes biochemical analysis on the DNA binding of human RAD51 protein involved in homologous recombination using surface plasmon resonance (SPR). The study particularly focuses on the effect of DNA flexibility (as well as single-stranded versus double-stranded) and of DNA length on association and dissociation rates of RAD51 to the DNA as well as polymerization rate on the DNA. The authors found that the flexibility of the DNA is a critical determinant of RAD51 polymerization on the DNA. The study also addressed the question how "rigid" RAD51 filaments are formed on "flexible" ssDNA by overcoming an entropic penalty and proposed enthalpic contribution by RAD51 polymerization would compensate the penalty. The works in the paper were carried out in a good care, the results are in a good quality and main conclusions are supported by the data. However, there seem more to support their conclusion in terms of biological significance of findings in the paper on RAD51-assembly in a cell for homology search and strand exchange. And also the authors need more words to readers in the journal, who are not familiar to this kind of biochemistry and modeling.

Major points:

1. Correlation between DNA flexibility (Table 1) and RAD51 polymerization rate (Fig. 3D) does not look strong. Particularly dA50 and dN-50p with different persistent length (4.7 and 7.3, respectively) show the similar polymerization rate (0.0005 and 0.0004, respectively). Thus, the polymerization does not seem to explain the different binding affinities of RAD51 to dA50 and dN-50p. More interesting, rather than wild type, RAD51-F86E mutant protein shows correlation between persistent length and polymerization rates (Fig. 4E), which suggests that rather than WT RAD51, RAD51-F86E seems to sense flexibility of DNAs.

We appreciate this reviewer's point. Nonetheless, given that WT RAD51 does exhibit significant differences in polymerisation rates between dT-50 and dN-50, and between these and both dA-50 and dN-50p, we are of the view that there is sufficient evidence to conclude that RAD51 polymerisation depends on persistence length.

That said, as the reviewer suggests, we appreciate that this relationship may not be linear. Interestingly, our data indicate that, beyond a persistence length of around ~4.7 nm, WT RAD51 exhibits a similar polymerisation rate. This is noteworthy as it is roughly on the same scale as a RAD51 molecule. Our model predicts that a single RAD51 can be added to an existing RAD51 polymer on DNA only when the persistence length of DNA is on the same scale as the size of a single RAD51 molecule so that ssDNA can bend out of the way. This nicely explains why the polymerisation rate (kp) for ssDNA dA and dN are much smaller and identical to the value for dsDNA (~0.004 / μ M/s). We have revised the text to reflect this more explicitly.

2. In the same line to #1 comment, as discussed by the authors (page 13, line 316-320), it would be great if the authors can perform SPR binding analysis of RAD51 protein using dN-50p with "double" tethering to the chip, which reduces the flexibility (entropy).

This is an interesting suggestion that we considered in detail since it would directly validate our model. The key issue is that, in SPR, the coupling of the oligos to the chip surface is random. This would then create a distribution of very rigid oligos (bound by a maximum stretch) to those that are very flexible (bound by minimum stretch). Accordingly, this approach would make interpretation of the results extremely challenging, with need to develop an additional layer of modeling that estimates the stretch distribution for each DNA molecule. We believe that the increased complexity would hinder, rather than assist, the interpretation of experimental readouts.

3. The authors proposed a bend-to-capture (BTC) mechanism to explain how the DNA flexibility affects polymerization. However, as authors pointed out in Figure 3A, the 2nd capture requires the dissociation of RAD51 from the 1st binding site. However, in their experiments, the authors used a condition in which RAD51 shows little dissociation from the DNA in the presence of Ca²⁺ (very long lifetime shown in Fig. 3E). How could very slow dissociation promote BTC? Thus, BTC does not seem to explain faster polymerization of RAD51 on more flexible DNAs. Or need more explanation in the text, if I am missing some points.

We apologise for the miscommunication on this matter. The BTC mechanism involves no step that requires dissociation of RAD51 from another RAD51 or from DNA. The figure may have led to this confusion; we have revised it so that a newly added RAD51 monomer is now marked with an asterisk and have revised the text to make this more explicit.

4. The authors proposed that enthalpic contribution of RAD51 polymerization is a key determinant to overcome entropic penalty on the flexible DNA. Is there an experimental evidence that RAD51 polymerization follows exothermic reaction by using isothermal titration calorimetry (ITC)?

As noted in the response to reviewer 1, it is not practical to perform ITC experiments for this specific reaction. We did not mean to make claims about the overall reaction entropy or enthalpy and therefore, we have revised the manuscript to explicitly focus on DNA flexibility and the RAD51 protomer-protomer interaction as contributing to the total free energy (without any claims about, for example, the overall enthalpy).

5. In Summary (page 2, line 34) and other parts (page 4, line 85), the authors used "mechano-sensor" for RAD51. What does this mean? The authors need more explanation on the

biological significance of mechano-sensor in homologous recombination. Since, as noted by the authors, ssDNAs produced in vivo are much longer than substrate used here and thus, intrinsically flexible or once coated with RPA, becomes more rigid. Without the additional information, it is premature to conclude that RAD51 is a mechano-sensor.

A mechano-sensor is typically defined as a biomolecule that responds to changes in mechanical force. In this context, we propose that RAD51 is a molecule that is able to form filaments on DNA according to the changes in DNA mechanical force (i.e., flexibility). As discussed in the manuscript, we agree that resected ssDNA is immediately bound by RPA, while also appreciating that RPA binding to ssDNA is dynamic. We propose that this dynamic property of RPA provides a temporary opening of hitherto occupied ssDNA for binding by five to ten molecules of RAD51. An excellent input on this aspect has been made by reviewer 1, which we have now incorporated in the discussion of the revised manuscript.

Minor points:

1. For DNA binding analysis by SPR, it would be nice to show the concentration effect on the binding curves to different types of DNAs. And also it would be nice to explain what Y-axis means. For example, in Fig. S1B, there is difference in SPR signal (shown in Y-axis) between ssDNA and dsDNA in the same length.

We are sorry about the confusion. Concentration impact has been taken into account in the modelling, and this point is now extensively explained in the revised manuscript. Y-axis is the average number of RAD51 molecules bound to a single DNA molecule, as explained in the method 'SPR Data Processing'. To avoid confusion, this information is now incorporated in the corresponding figure legends too.

2. One caveat in the modeling of Figure 2A and B etc, the authors do not seem assume that a single RAD51 multimer (Page 8, line 178-179); e.g. trimer binds to 5mer, which has a single RAD51 binding site (once RAD51 trimer binds to the 5mer, this triggers the dissociation of two RAD51?). Or this product might not show any dynamics. Consideration of these reactions should be important, since, under their condition, the authors indeed showed that RAD51 forms relatively long polymers whose average more than 10mer (site size should be 30 nucleotides) as shown in Fig. 2C and D.

While we appreciate this comment, we believe our assumption that a single RAD51 n-mer can only bind to a DNA molecule with at least n RAD51 binding sites is reasonable for following reasons. We observed no RAD51 binding to dN-5 (one RAD51 binding site), dN-8 (two RAD51 binding sites) and dN-11 (three RAD51 binding sites). This suggests that it is very unlikely that a RAD51 multimer of more than three molecules can bind to ssDNA molecules with three or fewer RAD51 binding sites. A similar concept applies to dN-5p, for which we observed no RAD51 binding. In regard to the binding observed with dN-14, dN-17, dN-8p and dN-11p, we cannot formally exclude that RAD51 n-mers were binding to these DNA molecules (RAD51 5-mers or larger to bind to dN-14, RAD51 6-mers or larger to bind to dN-17; RAD51 trimers or larger to bind to dN-8p; RAD51 tetramers or larger to bind to dN-11p). Adding these extra binding reactions would have made our model far more complex, but ultimately would lead us to similar conclusions. Applying these additional reactions would generate slightly lower polymerisation rate constant (k_p) values for all our data fitting, but this would apply to all our model fits, hence lead to the same trend in k_p values across dT-50, dN-50, dA-50 and dN-50p for both WT and F86E RAD51.

3. Page 5, line 101: WT RAD51 distinguishes ssDNA from dsDNA through faster polymerization on ssDNA. What data support this? Could be slop in the association phase. Need more word to readers.

We apologise for the confusion. This sentence refers to our initial observation shown in Figure 1, which agrees with the previous report by Candelli et al. The exact data of the polymerisation rates (k_p), identified by the SPR data-derived mathematical modelling, is provided in Figure 3A and 3B. We now provide the extensive descriptions and methodologies in the corresponding paragraphs in the revised manuscript.

4. Page 6, line 134: What is "ODE"?

We apologise for not denoting this abbreviation in the main text. This refers to 'ordinary differential equation', which is now inserted in the revised manuscript.

5. Page 8, line 184: 2.6+/-1.8 should be 2.6+/-"0.18".

We thank this reviewer for identifying the error. This is now corrected.

6. Page 11, line 254: Add (Fig. 4C) after dN-50p.

We thank this reviewer for identifying our omission. This is now inserted.

Referee #3:

The RAD51 proteins binds both single and double-stranded DNA. Therefore, it has been of considerable interest to understand mechanisms that allow RAD51 to preferentially bind to ssDNA, as only this binding permits homologous recombination.

Paoletti et al here base their experiments on previous findings that indicated that RAD51 polymerizes faster on single as opposed to double stranded DNA. To explain how RAD51 can discriminate between the two, they use a combination of SPR and mathematical modeling to suggest that flexibility of DNA plays an important role. In particular, they use various DNA sequences (dT, dA and scrambled sequence) that differ in terms of rigidity and find a correlation between polymerization rate and the flexibility/rigidity of the DNA sequences used. Likewise, dissociation of RAD51 from DNA is dependent on DNA rigidity. This is suppressed when using wt RAD51 mutant that stabilizes DNA, but is revealed when using a mutant with impaired RAD51-RAD51 interaction that renders the nucleoprotein filament less rigid.

Generally, the manuscript is well written and the experiments are well-explained. My main concern is whether the observations are correlative or causative. In other words, what if RAD51 prefers to bind dT, and it has little to do with DNA flexibility. In this regard, the used method has a limitation. It would be very nice to complement the results with an alternative method. I.e. molecular tweezers have been used previously to monitor assembly of RAD51 filaments, and the method seems to be much better suited to verify author's conclusions, as variation of applied force would directly affect DNA flexibility, without concern about DNA sequence difference.

Indeed, optical tweezers were previously exploited to assess assembly and disassembly of RAD51 filaments (van der Heijden T. et al., 2007; Candelli A et al.,2014), prompting us to conduct a pilot experiment at the LUMICKS (Netherlands) in Oct 2017 to explore if such an approach was feasible. Unfortunately, however, we found a number of limitations of using this approach to test our hypothesis. Specifically, the currently established molecular tweezer system uses a long DNA, often ~ 10 kb, which would allow multiple RAD51 nucleation events on the same molecule, possibly dependent on local flexibility, making the interpretation of the result less straightforward. Also, this system's measurement of RAD51 binding to DNA relies

on the force and length alteration of double-tethered DNA, exploiting the well-established phenomenon that RAD51 binding stretches DNA and increase rigidity. This means that RAD51 binding on a force-stretched DNA molecule is not quantifiable. The alternative method of using fluorescently-labelled RAD51 was also considered, but as the double-tethered DNA needs to be immersed in a channel full of labelled RAD51, fluorescence-based real-time detection of RAD51 binding to DNA was predicted to be challenging. We were also concerned about the scanning speed for detecting fluorescence, photobleaching and potential artefacts resulting from fluorescently labelling RAD51. It is worthy to note, however, that a recent optical tweezer-based study of RAD51 dissociation, involving transfer of the RAD51-DNA complex to a RAD51-free channel, demonstrated that tension has no impact in RAD51 dissociation (Brouwer I et al., 2018, Fig. 2), which is in agreement with our SPR-based model (Fig. 4E in this manuscript, defined as lifetime). This observation validates the confidence and accuracy of our assessment.

This study exploited a new SPR-based approach to test our hypothesis. Our SPR experiments are designed to limit experimental complexities and hence associated errors, and the real-time high-resolution readouts were directly fed into the mathematical modelling. The modelling also allowed us to take account of potential RAD51 multimer in solution according to its concentration, which to our knowledge had not been considered in previous studies.

Importantly, current structural studies have demonstrated that the presynaptic RAD51 filament engages the ssDNA backbone, exposing the nucleotide bases for homologous pairing reactions (Short JM et al., 2016; Xu J et al., 2017). Although insertions of unstructured RAD51 loops are found between bases, they are not shown to be involved in active interaction with DNA.

In summary, we conclude that RAD51 prefers flexible DNA rather than the base in dT-polymers. This point is now discussed in the manuscript.

My other comments are discussion points:

- The authors refer to the studies from the Rothstein and Gasser labs, which observed increased DNA mobility upon DNA breakage. This increased mobility was, however, clearly dependent on Rad51 and checkpoint, and it therefore cannot explain or facilitate RAD51 recruitment. The way the authors refer to these studies is somewhat misleading.

We apologise for the confusion. In this paragraph, we discuss the broader implications of this study for the recruitment of DNA-binding proteins in mammalian cells, beyond the specific role of RAD51. The cause and impact of damage-induced chromatin mobility in mammalian cells remain ill-defined. We trust that the manuscript outlines this point clearly.

- The authors propose that displacement of RPA will yield ssDNA of 30 nt in length, which might yield highly flexible DNA. It seems unlikely to me that the exposed DNA would be highly flexible, if it is surrounded on both ends with RPA-coated ssDNA of hundreds of nucleotides in length.

The discussion is further elaborated according to reviewer 1's input. Clearly, this is speculation, but we feel it is worth discussing as displacement of a single RPA molecule will microscopically increase the flexibility / bending capacity, which would be enough to facilitate the assembly of five to ten RAD51 molecules. Future studies in the presence of RPA, as well as other HR mediators, will test this hypothesis. This point is now discussed in the revised manuscript.

Thank you for submitting a revised version of your manuscript on RAD51 polymerization on flexible DNA for our consideration. All three original referees have now re-assessed it and found the study significantly improved and the previous concerns generally satisfied. We shall therefore be happy to accept the study for publication in The EMBO Journal, pending answering the remaining questions of referee 2 (without need for further experiments at this stage) during a final round of minor revision. You may simply send a response to the final comments and a modified version of the main text (with modifications highlighted via 'Track Changes' option) via e-mail.

REFEREE REPORTS

Referee #1 (Report for Author)

The authors adequately responded to my previous concerns.

While I would have loved to see the van't Hoff analysis (the change in the temperature does not need to be big), the authors have re-framed the discussion in a way that more consistent with their observations. The manuscript looks much stronger now. Overall this work will be a valuable contribution to our understanding of the RAD51 recombinase mechanism.

Referee #2 (Report for Author)

EMBOJ-2019-103002R, corresponding author Dr. Esashi
"Molecular flexibility of DNA as a key determinant of RAD51 recruitment"

A revised version of the paper by Paoletti et al. describes biochemical analyses on the DNA binding of human RAD51 protein involved in homologous recombination using surface plasmon resonance (SPR) and modeling. They showed DNA flexibility is a key determinant for RAD51-binding to DNAs. The authors addressed some (not all) my previous comments but rebutted the others.

The authors proposed bend-to-capture (BTC) mechanism to explain how the DNA flexibility affects polymerization, which was hard to follow in the previous version. Revised description of the model (Figure 4A) is much easier to follow compared to the previous version. Here is very naive question, on which the authors may explain again to this reviewer. It looks BTC could explain why RAD51 prefer flexible DNAs for its binding. On the other hand, that could not explain why RAD51 dislikes stiff DNAs. If binding energy for RAD51-RAD51 protomers is high (relative to RAD51-DNA binding in Figure 4A), RAD51 does not care the flexibility of the DNA, since a DNA binding site for incoming RAD51 is present (available) on the DNA bound to RAD51 polymer (I may misunderstand "steric clash" in Page 9 line 9; indeed what does it mean?). Thus, for BTC to work in Figure 4A, intrinsic DNA binding should be relatively stronger than the protomer interaction. Is this a case of RAD51?

Second, I thought that ITC is not a difficult experiment to perform (compared to optical-tweezer experiment), which would be very much helpful to support the authors' conclusion (EPC model) on enthalpic factor associated with RAD51-binding to DNA. Simple mixing of RAD51 with or without DNAs to see temperature change will tell the effect of DNAs (even though the interpretation may not be simple) on enthalpy.

Referee #3 (Report for Author)

This is a revised version of a manuscript. The authors have addressed the comments and explained why some of the proposed experiments were not possible. Overall, this is a very carefully done study that addresses a little explored mechanism that affects the formation of RAD51 filaments. I am happy to support it for acceptance.

Response to Referee #2

EMBOJ-2019-103002R, corresponding author Dr. Esashi
 "Molecular flexibility of DNA as a key determinant of RAD51 recruitment"

A revised version of the paper by Paoletti et al. describes biochemical analyses on the DNA binding of human RAD51 protein involved in homologous recombination using surface plasmon resonance (SPR) and modeling. They showed DNA flexibility is a key determinant for RAD51-binding to DNAs. The authors addressed some (not all) my previous comments but rebutted the others.

The authors proposed bend-to-capture (BTC) mechanism to explain how the DNA flexibility affects polymerization, which was hard to follow in the previous version. Revised description of the model (Figure 4A) is much easier to follow compared to the previous version. Here is very naive question, on which the authors may explain again to this reviewer. It looks BTC could explain why RAD51 prefer flexible DNAs for its binding. On the other hand, that could not explain why RAD51 dislikes stiff DNAs. If binding energy for RAD51-RAD51 protomers is high (relative to RAD51-DNA binding in Figure 4A), RAD51 does not care the flexibility of the DNA, since a DNA binding site for incoming RAD51 is present (available) on the DNA bound to RAD51 polymer (I may misunderstand "steric clash" in Page 9 line 9; indeed what does it mean?). Thus, for BTC to work in Figure 4A, intrinsic DNA binding should be relatively stronger than the protomer interaction. Is this a case of RAD51?

We are pleased that the referee found the revised Figure 4A easier to follow. This figure, however, clearly simplifies the real molecular interaction of RAD51-RAD51 protomers and RAD51-DNA. To be more precise, it would be best to refer to the RAD51-DNA complex structure shown by Xu et al., 2017.

Panel A shows a representative view of RAD51-DNA structure from the study (PDB: 5H1C, visualised by UCSF Chimera), where three RAD51 molecules assemble on dsDNA. The RAD51 molecule in the middle is coloured in pink, dsDNA is coloured in light blue, and ATP molecules are shown in yellow. Red lines indicate predicted clash/contact residues between RAD51-RAD51 and RAD51-dsDNA.

This structure shows that, in RAD51-DNA complex, (1) RAD51-RAD51 interaction involves three interfaces (i.e., one ATP-mediated primary interface and two RAD51-RAD51 stabilising interfaces), and (2) one RAD51 molecule binds to three phosphate groups in DNA backbone through three distinct interfaces. This suggests that for one RAD51 molecule to be incorporated into existing RAD51-DNA complex, six interfaces need to form. On a stiff DNA platform, it is not possible to simultaneously form these 6 interfaces without steric clashes. Panel B shows a simplified depiction of this, albeit more complex than the main Figure 4A, to explain this model. If DNA is able to bend, then steric clashes can be avoided because RAD51 can bind (or form an interface with) DNA first or RAD51 first before the second interface is formed. Therefore, stiff DNA introduces a penalty in the binding rate. A stronger RAD51-DNA interaction would not help this model, as it likely results in the misalignment of newly

incorporated RAD51 to the existing RAD51-DNA filament. We now clarify this in our re-revised manuscript.

Second, I thought that ITC is not a difficult experiment to perform (compared to optical-tweezer experiment), which would be very much helpful to support the authors' conclusion (EPC model) on enthalpic factor associated with RAD51-binding to DNA. Simple mixing of RAD51 with or without DNAs to see temperature change will tell the effect of DNAs (even though the interpretation may not be simple) on enthalpy.

As the referee 1 appreciates, the execution and interpretation of ITC experiment of human RAD51 is expected to be challenging. It is well known that human RAD51 forms polymers at high concentration even in the absence of DNA, as our mathematical model also predicts (Fig. 3C). While our SPR experiments were conducted at 150 nM of RAD51 for its binding to 50-mer DNA, ITC experiments require a higher concentration of proteins at 10 – 100 μ M (https://www.embl.fr/services/macromolecular_interactions/sample_guidelines/). This is far more than the physiological RAD51 concentration (100 nM). Moreover, unlike SPR that continuously injects RAD51 over the surface and therefore the polymer distribution of RAD51 in solution is constant over time, in the case of ITC, binding to DNA would deplete RAD51 in solution introducing additional dynamics that would need to be inferred from the data. We hope that, in the revised manuscript (v2), our model is clearly explained without claims about the overall enthalpy.

Corresponding Author Name: Prof. Fumiko Esashi & Dr. Omer Dushek

Journal Submitted to: The EMBO Journal

Manuscript Number: EMBOJ-2019-103002R